# Efficient Prediction of SO(3)-Equivariant Hamiltonian Matrices via SO(2) Local Frames

## Abstract

We consider the task of predicting Hamiltonian matrices to accelerate electronic structure calculations, which plays an important role in physics, chemistry, and materials science. Motivated by the inherent relationship between the off-diagonal blocks of the Hamiltonian matrix and the SO(2) local frame, we propose a novel and efficient network, called QHNetV2, that achieves global SO(3) equivariance without the costly SO(3) Clebsch–Gordan tensor products. This is achieved by introducing a set of new efficient and powerful SO(2)-equivariant operations and performing all off-diagonal feature updates and message passing within SO(2) local frames, thereby eliminating the need of SO(3) tensor products. Moreover, a continuous SO(2) tensor product is performed within the SO(2) local frame at each node to fuse node features. Extensive experiments on the large QH9 and MD17 datasets demonstrate that our model achieves superior performance across a wide range of molecular structures and trajectories, highlighting its strong generalization capability. The proposed SO(2) operations on SO(2) local frames offer a promising direction for scalable and symmetry-aware learning of electronic structures.

## 1 Introduction

Quantum Hamiltonian, as a central element in the many-body Schrödinger equation of quantum mechanics, plays a key role in governing the quantum states and physical properties of molecules and materials, making it essential for physics, chemistry, and materials science. First-principles methods such as density functional theory (DFT) (1; 2; 3) have been developed to solve the Schrödinger equation and investigate the electronic structures of molecules and solids. Despite their success, these methods struggle with high computational cost, limiting their application to systems with only a few hundred atoms. In DFT, the central Kohn-Sham equation is solved using the self-consistent field (SCF) method based on the variational principle of the second Hohenberg-Kohn theorem (2). The electronic wavefunctions, energies, charge density, and Kohn-Sham Hamiltonian are iteratively calculated and updated until convergence is achieved. This SCF calculation has a high time complexity of $O(N_e^3 T)$, where $N_e$ is the number of electrons and $T$ is the number of SCF steps required for convergence. Consequently, DFT remains computationally expensive for large and diverse quantum systems B.1.

Recently, deep learning has demonstrated great potential in advancing scientific research (4; 5; 6; 7). In particular, several machine learning models (8; 9; 10; 11; 12; 13; 14; 15) have been developed to directly predict the Hamiltonian matrix from atomic structures, achieving substantial speedup of several orders of magnitude in inference time compared to traditional DFT calculations. Despite these advances, achieving higher accuracy and more efficient training remains a key challenge in the development of quantum tensor networks for predicting Hamiltonian matrices and many other multi-physical coupling matrices.

To address the above challenge, here we propose a novel and efficient network QHNetV2, motivated by the inherent relationship between the off-diagonal blocks of the Hamiltonian matrix and the SO(2) local frame to achieve global SO(3) equivariance without using the computationally intensive SO(3) Clebsch–Gordan tensor products. Specifically, we first eliminate the need of SO(3) tensor products by introducing a set of new efficient and powerful SO(2)-equivariant operations and performing all off-diagonal feature updates and message passing within SO(2) local frames. Second, to effectively perform nonlinear node updates, we apply a continuous SO(2) tensor product within the SO(2) local frame at each node to fuse node features, mimicking the symmetric contraction module used in

MACE (16) for modeling many-body interactions. We conduct extensive experiments on the large QH9 and MD17 datasets, which shows the superior performance and strong generalization capability of our new framework over a diverse set of molecular structures and trajectories. Additionally, these novel SO(2) operations on SO(2) local frames presents a promising avenue for scalable and symmetry-aware learning of electronic structures.

## 2 PRELIMINARIES

### 2.1 HAMILTONIAN MATRICES AND SO(3) EQUIVARIANCE

The Kohn-Sham equation of molecular and materials systems is given by: $\hat{H}_{\mathrm{KS}}|\psi_n\rangle = \epsilon_n|\psi_n\rangle$, where $\hat{H}_{\mathrm{KS}}$ is the Kohn-Sham Hamiltonian operator, $\psi_n$ is single-electron eigen wavefunction (also called molecular orbital), and $\epsilon_n$ is the corresponding eigen energy. Under a predefined basis set such as the STO-3G atomic orbital basis $\{\phi_p\}$ combining radial functions with spherical harmonics, the Kohn-Sham equation can be converted into a matrix form: $\mathbf{HC} = \epsilon\mathbf{SC}$, with Hamiltonian matrix element $\mathbf{H}_{pq} = \int \phi_p^*(\mathbf{r})\hat{H}_{\mathrm{KS}}\phi_q(\mathbf{r})d\mathbf{r}$ and overlap matrix element $\mathbf{S}_{pq} = \int \phi_p^*(\mathbf{r})\phi_q(\mathbf{r})d\mathbf{r}$. $\epsilon$ is a diagonal matrix of the eigen energies, and $\mathbf{C}$ contains wavefunction coefficients, with each eigen wavefunction $\psi_n$ as a linear combination of the basis functions $\psi_n(\mathbf{r}) = \sum_p \mathbf{C}_{pn}\phi_p(\mathbf{r})$ (17). More details can be found in Appendices B.1, B.2.

Equivariance is a fundamental symmetry that must be preserved in the Hamiltonian matrix prediction task. It arises from representing Hamiltonian matrix and wavefunctions in specific basis sets such as atomic orbitals in DFT calculations which are sensitive to the spatial orientation. As a result, when a molecular system undergoes a global rotation characterized by Euler angles $(\alpha, \beta, \gamma)$, its Hamiltonian matrix must transform accordingly. Each atom pairs and their orbital pairs can be labeled as $(a_i, o_s, a_j, o_t)$. $a_i$ and $a_j$ are the atom indices which may refer to the same atom ($i = j$) or to different atoms ($i \neq j$) in the molecular system. $o_s$ and $o_t$ indicate the orbitals belonging to $a_i$ and $a_j$, respectively, with angular momentum quantum numbers $\ell_s$ and $\ell_t$ and magnetic quantum number $m_s$ and $m_t$. Under a rotation specified by Euler angles $(\alpha, \beta, \gamma)$, the corresponding Hamiltonian matrix block transforms as $\mathbf{H}'_{ij,st} = \left(D^{\ell_s}(\alpha, \beta, \gamma)\right)^{-1} \mathbf{H}_{ij,st} D^{\ell_t}(\alpha, \beta, \gamma)$, where two Wigner-D matrices $D^{\ell}(\alpha, \beta, \gamma)$ are applied to the left and right sides of the original Hamiltonian matrix block, respectively, according to the angular momentum of the orbital pairs in the block. This ensures the block-wise SO(3) equivariance of the Hamiltonian matrix under rotations.

When learning the Hamiltonian matrix using machine learning models, it is essential to incorporate high-degree equivariant features that align with the angular momentum quantum numbers $\ell$ of atomic orbitals. This is particularly important for accurately capturing orbitals with high angular momentum such as $d$-orbitals with $\ell = 2$, requiring the highest degree of SO(3) equivariant features with $L_{max} \geq 4$. Furthermore, the pairwise interactions are needed for predicting all Hamiltonian matrix blocks while maintaining the block-wise SO(3) equivariance.

Machine learning demonstrates its power for property predictions, force field development, and so on (18; 19; 20; 21; 22; 23; 24). And numerous models have been built upon tensor product (TP) have proven to be an effective approach (25; 26; 23; 27; 28; 29; 27; 16; 30; 31; 32; 33) with high-degree equivariant features. However, the computational complexity of TPs is $O(L_{\max}^6)$ which increases significantly with the maximum degree $L_{\max}$, posing a substantial challenge for tasks such as Hamiltonian matrix prediction where a high $L_{\max}$ is often required. A promising alternative is to replace full TPs by SO(2) convolutions, as proposed in eSCN (34). This approach demonstrates that the SO(2) Linear operation can be equivalent to SO(3) TP while reducing computational complexity to $O(L_{\max}^3)$. This raises an intriguing direction to explore more diverse SO(2)-based operations for modeling the Hamiltonian matrix more effectively and accurately, particularly in scenarios requiring a high $L_{\max}$.

### 2.2 SO(3) AND SO(2) IRREDUCIBLE REPRESENTATIONS

Under an arbitrary 3D rotation, the atomic orbital basis rotates accordingly. As a result, the Hamiltonian matrix element defined on the atomic orbital pairs transforms in an equivariant manner. It is therefore necessary to introduce irreducible representations (irreps) of the 3D rotation group, *i.e.*

SO(3) group. The SO(3) group consists of all 3D rotations, represented by $3 \times 3$ orthogonal matrices with determinant 1. For each SO(3) irrep with degree $\ell \in \mathbb{N}_0$, the corresponding representation space has dimension $2\ell + 1$, with spherical harmonics $\{Y_m^\ell(\theta, \phi)\}_{m=-\ell}^{\ell}$ serving as a natural basis. For each group element $g \in$ SO(3) (*i.e.* 3D rotation), the corresponding matrix representation for the $\ell$-th irrep is the Wigner D-matrix $D^\ell(g)$ of shape $(2\ell + 1) \times (2\ell + 1)$. Then, $D^\ell(g)$ is applied via matrix multiplication to perform the corresponding SO(3) rotation in the representation space. To develop more advanced operations within the SO(2) symmetry space, it is essential to introduce irreducible representations (irreps) of the SO(2) group. The SO(2) group describes 2D planar rotations, which can be interpreted as rotations around a fixed axis in 3D space (typically the z-axis). The irreps of SO(2) has dimension 2 for $m > 0$ and 1 for $m = 0$ or can be represented as complex values, with order $m \in \mathbb{N}_0$. A natural basis function for SO(2) irreps is the set of real circular harmonics, which take the form $B^m(\delta) = [\sin(m\delta), \cos(m\delta)]^T$ for $m \geq 1$ and $B^0(\delta) = [1]$ for $m = 0$. Each group element $g \in SO(2)$ corresponds to a rotation by an angle $\varphi \in [0, 2\pi)$, and the matrix representation for the $m$-th irrep is given by $\begin{pmatrix} \cos(m\varphi) & \sin(m\varphi) \\ -\sin(m\varphi) & \cos(m\varphi) \end{pmatrix}$ for $m > 0$, and 1 for $m = 0$. Subsequently, the representation space under SO(2) rotation transforms via matrix multiplication. In the complex circular harmonics basis $e^{im\delta}$, the representation space under SO(2) rotation transforms via multiplication with $e^{im\varphi}$.

## 3 METHODS

This section introduces the model we have built for the Hamiltonian matrix prediction task. The key is to develop the SO(2) local frame, where any powerful SO(2) equivariant operations can be applied while maintaining the overall framework to be SO(3) equivariant. Here, we would like to clarify that we adopt minimal frame averaging and extend the operation in eSCN to construct local frames, enabling the application of arbitrary SO(2) operations and supporting frame construction on nodes, edges, and node pairs beyond edges alone. Later on, we demonstrate that the model based on SO(2) local frame achieve great efficiency and accuracy, especially these two key things for building this networks especially considering the high $L_{\max}$ is an unavoidable things.

### 3.1 SO(2) LOCAL FRAMES

While the computation cost for SO(3) operations remains high, SO(2) operations can be more efficient to conduct. Besides the SO(2) linear operation, which has been verified in previous work (34) to maintain overall SO(3) equivariance, there is a need to provide a mechanism that supports the usage of arbitrary SO(2) operations while ensuring the framework remains SO(3)-equivariant. Therefore, inspired by the minimal frame averaging technique (35), we construct local frames that require SO(2) equivariance internally and employ canonicalization to guarantee overall SO(3) equivariance.

**Global Frame.** The overall framework is built on a global coordinate system together with local SO(2) frames. In the global system, the model operates on SO(3) irreps and preserves full SO(3) equivariance. Consequently, the input to each local network $\Phi$ is SO(3)-equivariant.

**Definition of the Local Frame.** A local frame is defined as a mapping from the 3D Euclidean space to SO(3):

$$\mathcal{F} : \mathbb{R}^3 \rightarrow \text{SO(3)}. \tag{1}$$

To make this concrete, we introduce a fixed target vector $\hat{v} \in \mathbb{R}^3$ that is used in the minimal frame construction. Given a unit direction vector $\hat{r} \in \mathbb{R}^3$, which rotates consistently with the input 3D data, and the fixed target vector $\hat{v}$, the local frame $F(\hat{r})$ is the rotation that maps $\hat{r}$ onto $\hat{v}$. For example, if we take the node pair direction $\hat{r}_{ij}$ and set $\hat{v} = (0, 0, 1)$, this reduces exactly to the setup used in eSCN. Thus, eSCN can be viewed as a special case of our more general framework. As proved in Appendix C, such a local frame can be applied to any SO(2) operation beyond the linear case by incorporating local frame averaging. Moreover, the placement of SO(2) local frames can be extended whenever a reference unit vector $\hat{r}$ is available, for example by defining a local frame on each node.

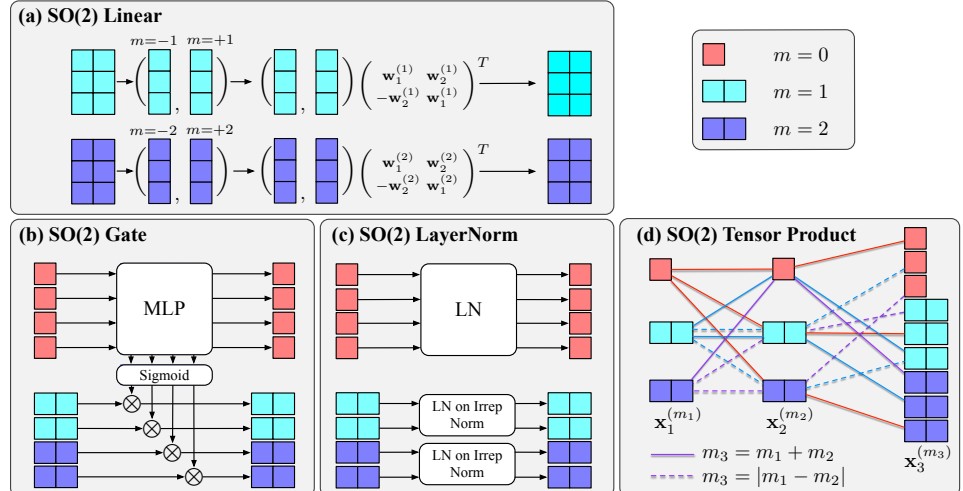

Figure 1: SO(2) equivariant operations. (a) SO(2) Linear. For SO(2) irreps with order $m > 0$, this operation uses weight matrices $\mathbf{w}_1^{(m)}, \mathbf{w}_2^{(m)} \in \mathbb{R}^{C \times C}$ where $C$ is the number of channels for input irreps. (b) SO(2) Gate. For the $m = 0$ features, a multi-layer perceptrons (MLP) is used to update them. Simultaneously, for each irrep with order $m > 0$, the MLP outputs a gate value passed through a sigmoid function, which modulates the corresponding SO(2)-equivariant features. (c) SO(2) LayerNorm (LN). For the $m = 0$ features, a standard LN is applied. For $m > 0$ features, LN is applied on the norm of SO(2) irreps according to Eq. 6. (d) SO(2) Tensor Product (TP). The SO(2) tensor product fuses features by combining irreps under the constraints $m_3 = m_1 + m_2$ (shown as solid lines) or $m_3 = |m_1 - m_2|$ (shown as dashed lines). The color of the path corresponds to its originating SO(2) irreps. A more general case containing $v - 1$ TPs for $v$ set of SO(2)-equivariant features is shown in Eq. 9. Each valid combination defines a path, ensuring the resulting features remain SO(2)-equivariant.

**Mapping Between** SO(3) **and** SO(2) **Irreps.** We use $FR(\hat{r})$ to denote the operation that projects SO(3) irreps into SO(2) irreps within the local frame, and $FR(\hat{r})^{-1}$ for the inverse mapping. Formally, given an SO(3) irrep $x \in \mathbb{R}^{2\ell+1}$ with components indexed by $m \in \{-\ell, \ldots, \ell\}$, the corresponding SO(2) irrep $x'$ after $FR(\hat{r})$ is defined as

$$x'_m = \sum_{m'=-\ell}^{\ell} D^\ell(R)_{m,m'} \, x_{m'}, \tag{2}$$

where $D^\ell(R) \in \mathbb{R}^{(2\ell+1) \times (2\ell+1)}$ is the Wigner-$D$ matrix associated with the rotation $R$ obtained from the canonicalization procedure to rotate reference direction $\hat{r}$ onto fixed target vector $\hat{v}$. The equivariance of such transformation is shown in Appendix E.

### 3.2 SO(2) Equivariant Operations

Based on the above analysis of SO(2) local frame, any SO(2)-equivariant layer can be applied within this frame while preserving the overall SO(3)-equivariance of the architecture. In this subsection, we discuss several SO(2) equivariant building blocks used within the SO(2) local frame.

**SO(2) Linear.** SO(2) Linear layer is first introduced in eSCN (34), and used in following works like EquiformerV2 (36). This linear operation takes SO(2) irreducible representations $\mathbf{x}$ as input, and applies the multiplication between the $\mathbf{x}$ and weights $\mathbf{w}$ with the formulation defined as

$$\begin{pmatrix} \mathbf{z}_{c,-m} \\ \mathbf{z}_{c,\ m} \end{pmatrix} = \sum_{c'} \begin{pmatrix} \mathbf{w}_{1,cc'}^{(m)} & \mathbf{w}_{2,cc'}^{(m)} \\ -\mathbf{w}_{2,cc'}^{(m)} & \mathbf{w}_{1,cc'}^{(m)} \end{pmatrix} \begin{pmatrix} \mathbf{x}_{c',-m} \\ \mathbf{x}_{c',\ m} \end{pmatrix}, \tag{3}$$

where $c'$ is the channel index for input $\mathbf{x}$ and $c$ is the channel index for output features $\mathbf{z}$. This linear operation can be easily understood if we convert them into complex numbers. Specifically, we use

$\tilde{\mathbf{x}}_{c'}^{(m)} = \mathbf{x}_{c',m} + i\mathbf{x}_{c',-m} = \bar{\mathbf{x}}_{c'}^{(m)} e^{i\theta_{x^{(m)},c'}}$ and weights $\tilde{\mathbf{w}}_{cc'}^{(m)} = \mathbf{w}_{1,cc'}^{(m)} + i\mathbf{w}_{2,cc'}^{(m)} = \bar{\mathbf{w}}_{cc'}^{(m)} e^{i\theta_{w^{(m)},cc'}}$.
The above equation is equivalent to

$$\tilde{\mathbf{z}}_c^{(m)} = \sum_{c'} \tilde{\mathbf{w}}_{cc'}^{(m)} \tilde{\mathbf{x}}_{c'}^{(m)} = \sum_{c'} \bar{\mathbf{x}}_{c'}^{(m)} \bar{\mathbf{w}}_{cc'}^{(m)} e^{i\left(\theta_{w^{(m)},cc'} + \theta_{x^{(m)},c'}\right)}, \tag{4}$$

where $\tilde{\mathbf{z}}_c^{(m)} = \mathbf{z}_{c,m} + i\mathbf{z}_{c,-m}$. It denotes a complex linear layer without bias term, which includes internal weights on the scale $\bar{\mathbf{w}}^{(m)}$ and rotation with angle $\theta_{w^{(m)}}$, as well as self-interaction across all input channels. Note that there is a set of weights for each individual $\mathbf{x}^m$ with $m > 0$.

**SO(2) Gate.** Gate activation is a useful component on various networks for SO(3) irreducible representations (28; 36), and this operation can also be applied to the SO(2) irreducible representations. Specifically, the corresponding formulation is shown as

$$\mathbf{z}^{(m)} = \begin{cases} \text{MLP}(\mathbf{x}^{m=0}) & , \text{if } m = 0, \\ \text{Sigmoid}(\text{MLP}(\mathbf{x}^{m=0})) \circ \mathbf{x}^{(m)} & , \text{if } m > 0, \end{cases} \tag{5}$$

An MLP is applied to learn the $m = 0$ features, and the other MLP takes the $m = 0$ features as the input, applies a sigmoid gate activation, then multiplies with the $m > 0$ features to control the irreducible features.

**SO(2) Layer Normalization.** Within the overall framework, it is often necessary to maintain a set of SO(2)-equivariant features to model the target quantities. In our task, for each off-diagonal block of the Hamiltonian matrix, it is required to model its corresponding SO(2) features. Since LayerNorm (37) is an important technique to stabilize the training procedure, we extend to apply it to SO(2) irreducible representations $\mathbf{x}^{(m)}$, defined as

$$\text{LN}(\mathbf{x}^{(m)}) = \frac{\mathbf{x}^{(m)}}{\text{norm}(\mathbf{x}^{(m)})} \circ \left(\frac{\text{norm}(\mathbf{x}^{(m)}) - \mu^{(m)}}{\sigma^{(m)}} \circ g^{(m)} + b^{(m)}\right), \tag{6}$$

where $\mu^{(m)} = \frac{1}{C} \sum_{i=0}^{C} \text{norm}(\mathbf{x}_i^{(m)})$, $\mu^{(m)} \in \mathbb{R}^{N \times 1 \times 1}$ is the mean of the norm across the channels and $\sigma^{(m)} = \sqrt{\frac{1}{C} \sum_{i=0}^{C} (\text{norm}(\mathbf{x}_i^{(m)}) - \mu^{(m)})^2}$, $\sigma^{(m)} \in \mathbb{R}^{N \times 1 \times 1}$ is the standard derivation of norm over the channels. After normalization, learnable affine parameters are introduced with a scale factor $g^{(m)} \in \mathbb{R}^{1 \times 1 \times C}$ and a bias term $b^{(m)} \in \mathbb{R}^{1 \times 1 \times C}$, which are used to rescale and recenter the scale of the SO(2) equivariant features with $m > 0$. We refer to it as norm-based Layer Normalization (LN), consistent with Eq. 6.

**SO(2) Tensor Product (TP).** Similar to the tensor product used for SO(3) features, the SO(2) tensor product provides a mechanism to fuse SO(2) features. Unlike previous operations that treat SO(2) irreps with each order $m$ separately, the SO(2) tensor product enables interactions between irreps with different order $m$. Specifically, the formulation is given by

$$\mathbf{z}^{(m_o)} = \mathbf{x}_1^{(m_1),+1} \otimes \mathbf{x}_2^{(m_2),+1} = \begin{pmatrix} \mathbf{x}_{1,-m_1}^{(m_1)} \mathbf{x}_{2,\ m_2}^{(m_2)} + \mathbf{x}_{1,\ m_1}^{(m_1)} \mathbf{x}_{2,-m_2}^{(m_2)} \\ \mathbf{x}_{1,\ m_1}^{(m_1)} \mathbf{x}_{2,\ m_2}^{(m_2)} - \mathbf{x}_{1,-m_1}^{(m_1)} \mathbf{x}_{2,-m_2}^{(m_2)} \end{pmatrix}, \tag{7}$$

where $m_o = m_1 + m_2$, and it is the multiplication between two complex number fusing SO(2) features from different $m$.

Similarly, when $m_o = m_1 - m_2$ with $m_1 > m_2$, the corresponding SO(2) TP can be formulated as

$$\mathbf{z}^{(m_o)} = \mathbf{x}_1^{(m_1),+1} \otimes \mathbf{x}_2^{(m_2),-1} = \begin{pmatrix} \mathbf{x}_{1,-m_1}^{(m_1)} \mathbf{x}_{2,\ m_2}^{(m_2)} - \mathbf{x}_{1,\ m_1}^{(m_1)} \mathbf{x}_{2,-m_2}^{(m_2)} \\ \mathbf{x}_{1,\ m_1}^{(m_1)} \mathbf{x}_{2,\ m_2}^{(m_2)} + \mathbf{x}_{1,-m_1}^{(m_1)} \mathbf{x}_{2,-m_2}^{(m_2)} \end{pmatrix} \tag{8}$$

Given the input SO(2) irreps with maximum order $M_{max}$, SO(2) TP includes all valid paths satisfying $m_o = m_1 + m_2$, $m_o = m_1 - m_2$, $m_1 > m_2$ or $m_o = m_2 - m_1$, $m_2 > m_1$, where $0 \leq m_1, m_2 \leq M_{max}$. And the number of paths is $O(M_{max}^2)$.

Moreover, inspired by the symmetric contraction module proposed in MACE (16) which uses the generalized CG to fuse multiple SO(3)-equivariant features after aggregating the node features to perform many-body interactions, a continuous SO(2) TP can be implemented in a similar way to

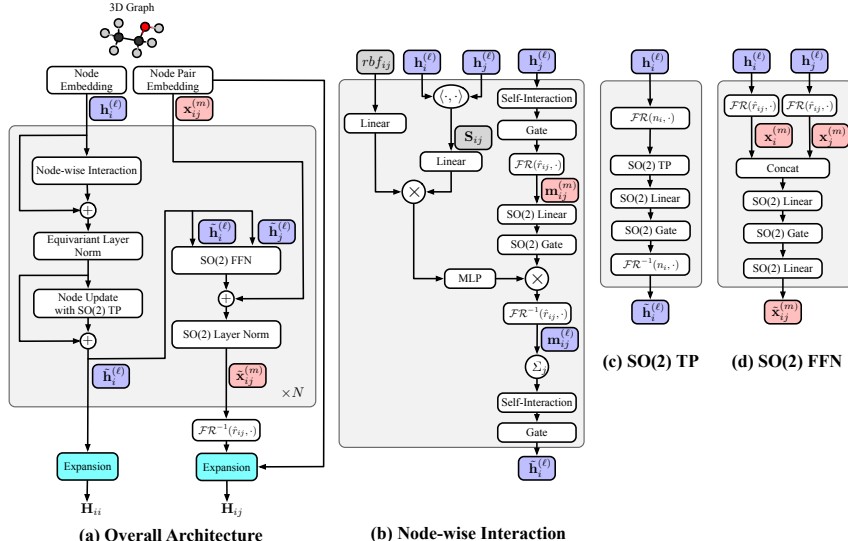

Figure 2: The overall architecture of the proposed QHNetV2. In this figure, $\times$ denotes element-wise multiplication, $\langle \cdot, \cdot \rangle$ denotes inner product. Gray color denotes scaler values, red color denotes SO(2) irreps, and blue color denotes SO(3) irreps.

capture more complex interactions. Specifically, we consider the interactions among $v$ sets of SO(2)-equivariant features and for each path with constraint $m_o = s_1 m_1 + s_2 m_2 + \cdots + s_v m_v$ where $s$ is the sign value within the range $\{-1, +1\}$, the corresponding output of this path is

$$\mathbf{z}^{m_o} = \underbrace{\mathbf{x}^{(m_1), s_1} \otimes \mathbf{x}^{(m_2), s_2} \otimes \cdots \otimes \mathbf{x}^{(m_v), s_v}}_{(v-1)\text{ TP}}. \tag{9}$$

Considering a set of SO(2) irreducible representations with maximum order $M_{max}$, the number of paths for SO(2) TP is $O(M_{max}^v)$. For example, the computational cost for simple SO(2) TP between two SO(2) irreps will be $O(M_{max}^2)$, where $M_{max}$ is the cutoff of the order on the SO(2) irreps. In our implementation, we set $M_{max} = L_{max}$, since the SO(2) irreps are transferred from the SO(3) irreps whose cutoff is $L_{max}$. To see why the time complexity is linear in $M_{max}$, consider the computation for a single channel. The SO(2) TP enumerates all valid index pairs $(m1, m2)$ that satisfy the SO(2) selection rule $m = m1 + m2$ and $m = |m_1 - m_2|$. Because both $m_1$ and $m_2$ range from 0 to $M_{max}$, the number of valid combinations is proportional to $M_{max}^2$. For each valid path, the computational work is $O(1)$, consisting of a single complex multiplication between two SO(2) feature components. Therefore, the total complexity will be $M^2$ for two SO(2) irreps as input of SO(2) TP. Note that the time complexity of rotating into the SO(2) local frame and rotating back is $O(L_{max}^3)$. This is because multiplying an irreducible representation of degree $\ell$ by the corresponding rotation matrix requires $\ell^2$ operations. Summing this cost over all degrees $\ell = 0, 1, \ldots, L_{max}$ yields a total complexity proportional to $\sum_{\ell=0}^{L_{max}} \ell^2 = O(L_{max}^3)$.

**Relationship between SO(2) operations and SO(3) operations.** We show the connections between SO(3) TP and SO(2) linear, SO(2) Gate and SO(3) Gate in Appendix D. We use SO(2) TP as a way to fuse SO(2) irreps and demonstrate its help in improving the performance in Table 4. Currently, we regard SO(2) TP and SO(3) TP as non-equivalent operations. While SO(2) TP offers lower computational complexity, it does not necessarily guarantee superior performance compared to SO(3) TP.

### 3.3 MODEL ARCHITECTURE

With the above SO(2) equivariant operations, here we introduce our model based on these SO(2) local frames and the global coordinate system for the Hamiltonian matrix prediction task.

**Node Embedding and Node Pair Embedding.** The node embedding learns one embedding for each atomic type. The node pair embedding takes the node embeddings for both nodes as well as the

pairwise distance, denoted as

$$\mathbf{x}_{ij}^{m=0} = \text{MLP}\left(\text{Linear}(\mathbf{s}_{ij}) \circ \text{Linear}(\text{rbf}(\bar{r}_{ij}))\right) \qquad (10)$$

**Node-wise interaction.** For the node-wise interaction module, the message passing paradigm of Graph Neural Networks (GNNs) (38) is applied to aggregate node features from their neighbors while preserving equivariance. As illustrated in Figure 2(b), a self-interaction layer and gate (25) operation is first applied to the input SO(3) irreps $\mathbf{h}^{(\ell)}$. After that, for each message, the $\mathbf{h}_j^{(\ell)}$ is rotated into its local SO(2) frame $\mathcal{F}(r_{ij})$ obtaining the SO(2) irreps $\mathbf{m}_{ij}^{(m)}$. Then, SO(2) Linear and SO(2) Gate operations are applied to the message $\mathbf{m}_{ij}^{(m)}$. The radial basis function (rbf), encoding the pairwise distance information, denoted as $rbf_{ij}$, is combined with the inner product of the equivariant features from both source and target nodes to multiply with $\tilde{\mathbf{m}}_{ij}$. Specifically, the inner product of node features is defined as

$$\mathbf{s}_{ij} = \left(\langle\mathbf{h}_i^{\ell=0}, \mathbf{h}_j^{\ell=0}\rangle\|\langle\mathbf{h}_i^{\ell=1}, \mathbf{h}_j^{\ell=1}\rangle\|\cdots\|\langle\mathbf{h}_i^{\ell=\ell_{max}}, \mathbf{h}_j^{\ell=\ell_{max}}\rangle\right), \qquad (11)$$

where $\langle\cdot,\cdot\rangle$ denotes the inner product and $\|$ represents vector concatenation across different SO(3) degrees. Then, the message is defined as

$$\tilde{\mathbf{m}}_{ij}^{(m)} = \text{SO(2)Linear}(\mathbf{m}_{ij}^{(m)}) \circ \text{MLP}\left(\text{Linear}(\mathbf{s}_{ij}) \circ \text{Linear}(\text{rbf}(\bar{r}_{ij}))\right). \qquad (12)$$

Subsequently, $\tilde{\mathbf{m}}_{ij}^{(m)}$ is rotated back to the global coordinate system to get message $\mathbf{m}_{ij}^{(\ell)}$ which is then aggregated together. A self-interaction with gate is applied on the aggregated node irreps.

**Node feature updating with SO(2) Tensor Product (TP).** Motivated by the symmetric contraction module proposed in MACE, a node updating module is applied with consecutive SO(2) TPs after the aggregations, as shown in Figure 2(c).

First, we need to find the reference unit vector for each node to build the local frame on nodes. Specifically, for node $n_i$, we select the direction vector from the closest neighbor node to the center node to build the local SO(2) frame, denoted as

$$\mathcal{F}(n_i) = \mathcal{F}(\hat{r}_{ij}) \text{ with } \arg\min_{j \in \mathcal{N}_i} \bar{r}_{ij}, \qquad (13)$$

where $\bar{r}_{ij}$ is the pairwise distance. Then, the node feature updating module is performed as

$$\mathbf{x}_i^{(m)} = \mathcal{FR}\left(n_i, \mathbf{h}_i^{(\ell)}\right), \tilde{\mathbf{x}}_i^{(m)} = \text{SO(2)Linear}\left(\text{SO(2)TP}\left(\mathbf{x}_i^{(m)}, v\right)\right), \tilde{\mathbf{h}}_i^{(\ell)} = \mathcal{FR}^{-1}\left(n_i, \tilde{\mathbf{x}}_i^{(m)}\right), \qquad (14)$$

where the SO(2)TP operation collects the SO(2) irreps according to Eq. 9 with $v-1$ consecutive SO(2) TP operations. The collected SO(2) irreps are then fed into a SO(2) Linear before being transformed back to the global coordinate system. We show in Table 4 that with the SO(2) TP within the local frames on nodes can improve the performance.

Although selecting the nearest neighbor provides a fast and simple way to determine the local reference vector for each node, this approach can lead to discontinuities in frame construction (39). Previous work (40) addressed this issue by constructing $O(n^3)$ frames to ensure global continuity. Further improvements can be made by averaging over all local frames derived from the directions between each node and its neighbors, rather than selecting only a single frame per node as in the current framework. This approach requires building $O(n)$ local frames per atom, resulting in $O(n^2)$ frames in total.

**Off-diagonal feature updating with SO(2) Feed Forward Networks (FFNs).** The off-diagonal features account for the majority of the final Hamiltonian matrix, and there is a one-one correspondence between the SO(2) local frame and its off-diagonal matrix block. Therefore, we keep all the features within this SO(2) local frame over the layers.

For each layer shown in Figure 2(d), the features coming from the neighbors $\mathbf{h}_i^{(\ell)}$ and $\mathbf{h}_j^{(\ell)}$ are transferred to the local frame $\mathcal{F}(\mathbf{r}_{ij})$, obtaining the corresponding SO(2) irreps $\mathbf{m}_i^{(m)}$ and $\mathbf{m}_j^{(m)}$. Hence, the associated off-diagonal feature is given by

$$\mathbf{x}_{ij}^{(m)} = \text{SO(2) Linear}\left(\text{SO(2) Gate}\left(\text{SO(2) Linear}(\mathbf{m}_i^{(m)}\|\mathbf{m}_j^{(m)})\right)\right) \qquad (15)$$

**Overall architecture.** The overall architecture is demonstrated in Figure 2(a). It contains two parts. The left one updates the node features $\mathbf{h}_i^{(\ell)}$. The skip connection is applied after the node-wise interaction, and then, the Equivariant LayerNorm (36) is applied before feeding into the SO(2) TP operations. The right one is used to update the pairwise features $\mathbf{x}_{ij}^{(m)}$. With the updated pairwise node features $\tilde{\mathbf{h}}_i$ and $\tilde{\mathbf{h}}_j$, it applies the SO(2) FFN on them, followed by a skip connection with SO(2) LayerNorm to update the pairwise SO(2) irreps $\tilde{\mathbf{x}}_{ij}^{(m)}$.

**Matrix Construction.** We follow the expansion module in QHNet (10), which uses the $\left(\bar{\otimes} w^{\ell_3}\right)_{(m_1,m_2)}^{(\ell_1,\ell_2)} = \sum_{m_3=-\ell_3}^{\ell_3} C_{(\ell_3,m_3)}^{(\ell_1,m_1),(\ell_2,m_2)} w_{m_3}^{\ell_3}$ and establishes a mapping between the irrep blocks and the orbital pairs within the Hamiltonian matrix blocks. For the diagonal matrix block, the node feature $\tilde{\mathbf{h}}^{(\ell)}$ is directly used as the input of Expansion module. For the off-diagonal matrix blocks, the SO(2) irreps are first transformed back to the global coordinate system and then fed into the Expansion module.

# 4 RELATED WORKS

Since intrinsic symmetries present in physical systems play an important role in modeling, equivariant neural networks (41) are explicitly designed to encode these symmetries directly into the architecture. By construction, these models preserve equivariant features throughout all layers, ensuring that symmetry-consistent representations are maintained. For Cartesian equivariant models such as PaiNN (42), ViSNet (43), and GotenNet (41), the architectures rely on scalars and vectors features within the model. These model are both efficient and powerful, enabling the models to achieve strong performance across a wide range of tasks. In contrast, spherical equivariant models are built on SO(3) irreducible representations and make extensive use of spherical harmonics and tensor products to rigorously encode rotational symmetries. Examples include TFN (25), SEGNN (23), SE(3)-Transformers (26), NequIP (28), MACE (16), Allegro (27) and Equiformer (29). These models build their architectures around tensor products as the central mechanism for encoding directional information within the irreducible representations of the feature space. Although tensor products offer a powerful mechanism for encoding geometric information, their computational cost has been widely discussed as a major bottleneck with a computational complexity $O(L^6)$, where $L$ is the maximum degree of the input irreducible representations. To improve computational efficiency, ESCN (34) proposes to reduce the SO(3) tensor product into SO(2) in this case that the SO(3) tensor product can be equivalently replaced by a SO(2) Linear. This approximation reduces the complexity to $O(L^3)$, enabling substantially faster training and inference while preserving essential symmetry properties, as demonstrated in EquiformerV2 (36). Motivated by the efficiency gains achieved through SO(2)-based operations, in this work we further generalize the SO(2) operations beyond SO(2)-linear layers. We show that any SO(2) operations can be incorporated within these SO(2) local frames while maintaining global SO(3) equivariance, enabled by a minimal frame-averaging mechanism. In addition, these SO(2) local frames can be constructed not only on edges but also on nodes or other locations where a well-defined directional reference can be established.

Numerous studies have explored the task of Hamiltonian matrix prediction in quantum systems (11; 44; 45).SchNorb (46) takes the invariant SchNet (18) as its backbone and build models for directly predicting the Hamiltonian matrix blocks. Subsequent models such as PhiSNet (8) and QHNet (10) incorporate equivariance into Hamiltonian modeling by leveraging tensor product (TP) operations and equivariant neural networks to respect the underlying rotational symmetries of the system. SPHNet (14) introduces sparse gate on tensor product mechanism to accelerate equivariant computations while maintaining competitive performance. WANet (13) addresses the nonlinearity of the Hamiltonian's eigenvalues—*i.e.*, the energy spectrum—by introducing an auxiliary loss function that directly constrains the predicted eigenvalues, enhancing both accuracy and physical consistency. For materials datasets, the DeepH series of works (47; 48; 49; 50; 51) framework demonstrates the evolution from invariant to fully equivariant architectures, capable of predicting both Hamiltonians and higher-order tensors along material trajectories. HamGNN (12) takes use of TP to build the equivariant graph networks for its prediction, and DeePTB (15) designs a strictly local model that efficiently captures many-body interactions. As a plug-in module, TraceGrad (52) introduces an auxiliary objective based on the norm of the predicted Hamiltonian blocks and corresponding blocks for improving the performance of Hamiltonian matrix prediction. For the self-consistent training

Table 1: Performance on QH9 dataset. The unit for the Hamiltonian $\mathbf{H}$ and eigen energies $\epsilon$ is Hartree denoted by $E_h$. Lower is better for $\mathbf{H}$ and $\epsilon$; higher is better for $\psi$. The best performance scores are highlighted in **bold**. Underline indicates the second best performance scores.

| Dataset | Model | $\mathbf{H}\left[10^{-6}E_h\right]\downarrow$ | | | $\epsilon\left[10^{-6}E_h\right]\downarrow$ | $\psi\left[10^{-2}\right]\uparrow$ |
| | | diagonal | off-diagonal | all | | |
|---|---|---|---|---|---|---|
| QH9-stable-id | QHNet | 111.21 | 73.68 | 76.31 | 798.51 | 95.85 |
| | WANet | – | – | 79.99 | 833.61 | 96.86 |
| | SPHNet | – | – | 45.48 | **334.28** | 97.75 |
| | QHNetV2 | **73.62** | **28.30** | **31.50** | 417.89 | **98.58** |
| QH9-stable-ood | QHNet | 111.72 | 69.88 | 72.11 | 644.17 | 93.68 |
| | SPHNet | – | – | 43.33 | 186.40 | **98.16** |
| | QHNetV2 | **61.09** | **20.81** | **22.97** | **165.89** | 97.68 |
| QH9-dynamic-300k-geo | QHNet | 166.99 | 95.25 | 100.19 | 843.14 | 94.95 |
| | QHNetV2 | **87.98** | **31.67** | **35.60** | **270.02** | **98.77** |
| QH9-dynamic-300k-mol | QHNet | 261.63 | 108.70 | 119.66 | 2178.15 | 90.72 |
| | QHNetV2 | **138.26** | **42.06** | **49.01** | **629.64** | **97.43** |

framework (53; 54; 55), it combines the machine models with a self-consistent field (SCF) loop during training, enabling unsupervised learning of quantum properties. Although this method enhances accuracy and expressiveness, it introduces significant computational overhead due to iterative refinement.

## 5 EXPERIMENTS

We evaluate and benchmark our model on QH9 in Section 5.1 and MD17 in Section 5.2. Moreover, we provide the efficiency studies in Appendix A.1 and ablation studies in Appendix A.2 of SO(2) TP and SO(2) FFNs, demonstrating the contributions of each proposed component. We notice that a new dataset, PubChemQH (13), has recently been introduced with larger molecule size and orbital size. However, as this dataset is not yet publicly available, we will leave the experiments on this dataset until access becomes open. Our code implementation is based on PyTorch 2.4.1 (56), PyTorch Geometric 2.6.1 (57), and e3nn (58). In experiments, we train models on 80GB Nvidia A100 with Intel(R) Xeon(R) Gold 6258R CPU @ 2.70GHz or 46GB NVIDIA RTX 6000 with AMD EPYC 9684X 96-Core Processor for QH9 and 11GB Nvidia GeForce RTX 2080Ti GPU with Intel Xeon Gold 6248 CPU for MD17.

**Evaluation metrics.** The evaluation metrics are the mean absolute error (MAE) on Hamiltonian matrix $\mathbf{H}$, as well as the eigen energies $\epsilon$ and cosine similarity on the wavefunction coefficients denoted as $\psi$ on the occupied orbitals.

### 5.1 QH9

**Datasets.** The QH9 dataset (11) (CC BY-NC SA 4.0 license) consists of four distinct tasks. The QH9-stable includes 130k molecules derived from QM9 (59; 60). The detailed description of QH9 dataset can be found in Appendix F.1.

**Training Details.** Our model follows the same training settings as the QH9 benchmark, with the corresponding hyperparameters provided in Table 6 in the Appendix F.1.

**Results.** The experimental results are presented in Table 1, where we can observe a clear improvement in the MAE of the Hamiltonian matrix $\mathbf{H}$. Compared to SPHNet, our model achieves a reduction in MAE on $\mathbf{H}$ by 33.7% on QH9-stable-id and 46.9% on QH9-stable-ood. The improvement is even larger when compared to QHNet. For the MAE of the $\epsilon$, our model achieves at least 47.6% error reduction compared to QHNet, and reasonable results compared to SPHNet.

### 5.2 MD17

**Datasets.** The MD17 datasets (46) (CC BY-NC license) consists of four molecular trajectories for water, ethanol, malondialdehyde, and uracil, respectively. The corresponding number of geometries for each trajectory for train/val/test split is provided in Table 7.

**Training Details.** The training hyperparameters are shown in Table 8 following the settings and split of previous works. We follow the experimental settings of QHNet on MD17, and the experimental

Table 2: Performance on MD17 dataset. The unit for the Hamiltonian $\mathbf{H}$ and eigen energies $\epsilon$ is Hartree denoted by $E_h$. Lower is better for $\mathbf{H}$ and $\epsilon$; higher is better for $\psi$. The best performance scores are highlighted in **bold**. Underline indicates the second best performance scores.

| Dataset | Method | Training Strategies | $\mathbf{H}[10^{-6}E_h]\downarrow$ | $\epsilon[10^{-6}E_h]\downarrow$ | $\psi[10^{-2}]\uparrow$ |
|---|---|---|---|---|---|
| Water | PhiSNet | LSW (10,000, 200,000) | 17.59 | 85.53 | 100.00 |
| | QHNet | LSW (10,000, 200,000) | **10.36** | **36.21** | 99.99 |
| | SPHNet | LSW (10,000, 200,000) | 23.18 | 182.29 | 100.0 |
| | QHNetV2 | LSW (10,000, 200,000) | 22.55 | 106.64 | 99.99 |
| Ethanol | PhiSNet | LSW (10,000, 200,000) | 20.09 | 102.04 | 99.81 |
| | QHNet | LSW (10,000, 200,000) | 20.91 | 81.03 | 99.99 |
| | SPHNet | LSW (10,000, 200,000) | 21.02 | 82.30 | 100.00 |
| | QHNetV2 | LSW (10,000, 200,000) | **12.05** | **70.46** | 99.99 |
| Malondialdehyde | PhiSNet | LSW (10,000, 200,000) | 21.31 | 100.60 | 99.89 |
| | QHNet | LSW (10,000, 200,000) | 21.52 | 82.12 | 99.92 |
| | SPHNet | LSW (10,000, 200,000) | 20.67 | 95.77 | 99.99 |
| | QHNetV2 | LSW (10,000, 200,000) | **10.85** | **67.46** | 99.92 |
| Uracil | PhiSNet | LSW (10,000, 200,000) | 18.65 | 143.36 | 99.86 |
| | QHNet | LSW (10,000, 200,000) | 20.12 | 113.44 | 99.89 |
| | SPHNet | LSW (10,000, 200,000) | 19.36 | 118.21 | 99.99 |
| | QHNetV2 | LSW (10,000, 200,000) | **10.38** | **107.42** | 99.91 |

results of QHNet and PhiSNet are from Table 1 in QHNet experiments. Since increasing the number of iterations can improve the final performance, to make a fair comparison, the total number of iterations are fixed for these experiments with a total 200,000 iterations. The original implementation and experiments of PhiSNet follows the settings of using reduce on the Plateau and won't stop until the learning rate is reduced to or achieving a total number of iterations. This will lead to a very large number of iterations for training, and the number of iterations are not fixed and pretty large for all the models.

**Results.** The results of MD17 is summarized in Table 2. We can find that in ethanol, malondialdehyde and uracil, our model provides a significant better performance on the MAE for $\mathbf{H}$ with at least 42% error reduction. Meanwhile, our model provides a better MAE on $\epsilon$. For the water dataset, as shown in Table 7, it contains only 500 geometries compared to 25,000 geometries for each of the other three molecules. It indicates that our model tends to achieve better performance when the training set includes a larger number of geometries.

## 6 CONCLUSION

In this work, we propose a novel network QHNetV2 for Hamiltonian matrix prediction that leverages the SO(2) local frame to achieve global SO(3) equivariance while eliminating the need for tensor product (TP) operations in both diagonal and off-diagonal components. By introducing the SO(2) local frame, we develop a set of new SO(2)-equivariant operations to construct powerful and efficient neural networks. The proposed approach ensures high computational efficiency by avoiding costly SO(3) TPs entirely. Experimental results on the QH9 and MD17 datasets demonstrate the effectiveness of our method. Our work opens the door to further incorporating efficient and powerful SO(2)-based operations and frames into geometric deep learning, while preserving global SO(3) equivariance.

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

Table 3: Efficiency Studies. This efficiency comparison is performed on the QH9-stable-id task.

| Model | Memory [GB/Sample] ↓ | Speed [Sample/Sec] ↑ | Speedup Ratio ↑ |
|---|---|---|---|
| QHNet | 0.70 | 19.20 | 1.00x |
| SPHNET | **0.23** | 76.80 | 4.00x |
| QHNetV2 | 0.32 | **83.33** | **4.34x** |

Table 4: Ablation study results on the QH9-stable-id task for our model with SO(2) TPs on the node updating modules and SO(2) FFNs on the off-diagonal feature updating modules.

| Index | Architecture | | $\mathbf{H} \left[10^{-6}E_h\right] \downarrow$ | $\boldsymbol{\epsilon} \left[10^{-6}E_h\right] \downarrow$ | $\psi[10^{-2}] \uparrow$ |
|---|---|---|---|---|---|
| | SO(2) TP | SO(2) FFNs | | | |
| 1 | ✓ | ✓ | 31.50 | 417.89 | 98.58 |
| 2 | ✗ | ✓ | 36.50 | 429.17 | 98.18 |
| 3 | ✗ | ✗ | 49.84 | 823.93 | 97.88 |

# A  EXPERIMENTS

## A.1  EFFICIENCY STUDIES

To compare the efficiency of our model compared to existing baselines, we follow the settings of (14) to run our model on a single A6000 with the maximum available batch size, and compare the corresponding results are shown in Table 3. We can observe a clear speed improvement compared to previous QHNet model with 4.34x faster, demonstrating the efficiency improvement introduced by the new SO(2) operations to eliminate the SO(3) TP operations. Compared to the SPHNet which is based on sparse gate to prune the path in tensor product for accelerating the training procedure, our model shows a faster training speed with a slightly higher GPU memory occupation.

## A.2  ABLATION STUDIES

We performed ablation studies on two main components of our model: SO(2) TP and SO(2) FFN. The SO(2) TP is applied to update the node features, while the SO(2) FFN is applied to update the pair node features. Note that we compare the experimental results of adding these components and simply removing these components (the blocks are skipped). The final results are shown in Table 4. We observe that both proposed SO(2) TP and SO(2) FFNs improve the final model performance.

# B  BACKGROUND

## B.1  DENSITY FUNCTIONAL THEORY

Density Functional Theory (DFT) (1; 2; 3) provides a powerful and efficient computational framework for modeling electronic structures of quantum systems and predicting a wide range of chemical and physical properties for gas-phase molecules and solid-state materials. DFT-based electronic structure methods reduce the reliance on the laboratory experiments, significantly advancing scientific research in physics, chemistry, and materials science. The key motivation behind DFT is to address the well-known many-body Schrödinger equation that governs the quantum states of systems such as their energies and wavefunctions. Mathematically, the Schrödinger equation for a system of total $N_e$ electrons is given by: $\hat{H}\Psi\left(\mathbf{r}_1, \mathbf{r}_2, \cdots, \mathbf{r}_{N_e}\right) = E\Psi\left(\mathbf{r}_1, \mathbf{r}_2, \cdots, \mathbf{r}_{N_e}\right)$, where $\hat{H}$ is the Hamiltonian operator and $\Psi\left(\mathbf{r}_1, \mathbf{r}_2, \cdots, \mathbf{r}_{N_e}\right)$ is the many-body electronic wavefunction. Although the Schrödinger equation can describe the entire system exactly, its computational complexity grows exponentially with the number of electrons. Specifically, even without accounting for the spin degrees of freedom, the many-body electronic wave function $\Psi$ depends on $3N_e$-dimensional spatial variables. Hence, the associated function space expands exponentially with respect to $N_e$, making it computationally infeasible for computers to accurately solve complex and diverse quantum systems. To address this challenge, Hohenberg and Kohn proved that the ground-state properties of a many-body electronic system is uniquely determined by its three-dimensional electron density $\rho(\mathbf{r})$ (1), thus avoiding the need to explicitly handle the full $3N_e$-dimensional many-electron wavefunction

$\Psi(\mathbf{r}_1, \mathbf{r}_2, \cdots, \mathbf{r}_{N_e})$. Kohn and Sham proposed a practical approach to map a many-body interacting system onto a set of non-interacting one-body systems where each electron moves in an effective potential arising from the nuclei and the average effect of other electrons (2). This leads to the well-known single-particle Kohn-Sham equation: $\hat{H}_{\mathrm{KS}}|\psi_p\rangle = \epsilon_p|\psi_p\rangle$ for individual electrons, where $\hat{H}_{\mathrm{KS}}$ is the Kohn-Sham Hamiltonian, $\psi_p$ is the single-electron eigen wavefunction, and $\epsilon_p$ is the corresponding eigen energy. The Kohn-Sham Hamiltonian is given by $\hat{H}_{\mathrm{KS}} = \hat{T} + \hat{V}_{\mathrm{Hartree}} + \hat{V}_{\mathrm{XC}} + \hat{V}_{\mathrm{ext}}$, where $\hat{T} = -\frac{\hbar^2}{2m_e}\nabla^2$ is the kinetic energy operator, $\hat{V}_{\mathrm{Hartree}}$ is the Hartree potential, $\hat{V}_{\mathrm{XC}}$ is the exchange-correlation potential, and $\hat{V}_{\mathrm{ext}}$ is external potential. The electron density from the Kohn-Sham equation with the exact exchange-correlation energy functional, should be same as that of the many-body interacting system: $\rho(\mathbf{r}) = \sum_p \int f_p|\psi_p(\mathbf{r})|^2 d\mathbf{r} = \int |\Psi(\mathbf{r}, \mathbf{r}_2, \cdots, \mathbf{r}_{N_e})|^2 d\mathbf{r}_2 \cdots d\mathbf{r}_{N_e}$, where $f_p$ is the occupation number of the $p$-th Kohn-Sham eigen state. Therefore, solving the full many-electron wavefunction $\Psi$ is no longer required.

### B.2 ATOMIC ORBITALS AND HAMILTONIAN MATRICES

The Kohn-Sham equation, as the central equation in DFT, can be solved in a predefined basis set, such as the STO-3G atomic orbital basis which combines radial functions with spherical harmonics centered on each atom. The calculated Kohn-Sham eigen wavefunction $\psi_n$, often referred to as molecular orbital in the context of molecules, can be expressed as a linear combination of these predefined basis functions $\psi_n(\mathbf{r}) = \sum_p \mathbf{C}_{pn}\phi_p(\mathbf{r})$, known as the Linear Combination of Atomic Orbital (LCAO) method (17). Each single-electron wavefunction $\psi_n(\mathbf{r})$ depends only on three spatial variables, thereby avoiding the exponential scaling associated with the $3N_e$-dimensional full many-electron wavefunction. By applying the predefined orbitals and the LCAO method within DFT, the Kohn-Sham equation can be converted into the following matrix form:

$$\mathbf{HC} = \boldsymbol{\epsilon}\mathbf{SC}, \tag{16}$$

where $\mathbf{H}$ is the Hamiltonian matrix of size $\mathbb{R}^{N_o \times N_o}$. Each matrix element is defined by evaluating the Hamiltonian operator in a pair of predefined atomic orbitals, shown as $\mathbf{H}_{pq} \equiv \langle\phi_p|\hat{H}_{\mathrm{KS}}|\phi_q\rangle = \int \phi_p^*(\mathbf{r})\hat{H}_{\mathrm{KS}}\phi_q(\mathbf{r})d\mathbf{r}$. Here, $N_o$ denotes the number of predefined orbitals which typically increases linearly with the number of electronics $N_e$. $\mathbf{C} \in \mathbb{R}^{N_o \times N_o}$ denotes the wavefunction coefficient matrix. $\boldsymbol{\epsilon} \in \mathbb{R}^{N_o \times N_o}$ is a diagonal matrix where each diagonal element corresponds to the eigen energy of each molecular orbital. $\mathbf{S} \in \mathbb{R}^{N_o \times N_o}$ is the overlap matrix, where each entry is the integral of a pair of predefined basis over the spatial space $\mathbf{S}_{pq} = \int \phi_p^*(\mathbf{r})\phi_q(\mathbf{r})d\mathbf{r}$.

## C SO(3) EQUIVARIANCE OF SO(2) LOCAL FRAMES

Frame averaging (61; 39; 35) is a technique that enforces equivariance to any model $\Phi$ by an SO(3)-equivariant frame $\mathcal{F} : \mathbb{R}^3 \to \mathrm{SO}(3)$ such that

$$\langle\Phi\rangle_{\mathcal{F}}(\hat{r}) = \frac{1}{|\mathcal{F}(\hat{r})|} \sum_{g \in \mathcal{F}(\hat{r})} g \cdot \Phi\left(g^{-1} \cdot \hat{r}\right) \tag{17}$$

With a little abuse of the definition, we further consider the input space includes the SO(3) irreps which is build upon the global coordinate system. Since there are many candidate frame constructions for frame averaging, we use the minimal frame averaging (35) to give an analytic frame construction with a minimal frame size. To apply the minimal frame averaging (35), the canonicalization is required to transfer the input features into its canonical form, defined in Definition 3.3 (35). We define the canonicalization operation based on a reference vector $\hat{r}$ as $c(\hat{r}) = h^{-1} \cdot \hat{r} = \hat{v}$, where a rotation $h \in \mathrm{SO}(3)$ is applied to align the reference vector $\hat{r}$ with the fixed target vector $\hat{v}$.

By Theorem 3.2 in (35), the minimal frame is defined by $\mathcal{F}(\hat{r}) = h\mathrm{Stab}_{\mathrm{SO}(3)}(\hat{v})$ where

$$\mathrm{Stab}_{\mathrm{SO}(3)}(\hat{v}) = \{g \in \mathrm{SO}(3) \mid \mathrm{g} \cdot \hat{\mathrm{v}} = \hat{\mathrm{v}}\}.$$

Instead of employing an arbitrary local model which could make the direct sum over all $g \in \mathrm{Stab}_{\mathrm{SO}(3)}(\hat{v})$ computationally expensive or even intractable, since $\mathrm{Stab}_{\mathrm{SO}(3)}(\hat{v})$ is isomorphic to

SO(2) (rotations about the y-axis), we consider $\Phi$ to be $\text{Stab}_{\text{SO}(3)}(\hat{v})$-equivariant with respect to the reference axis $r$. That's to say, for any rotation $g \in \text{Stab}_{\text{SO}(3)}(\hat{v})$, the local network satisfies $\Phi(g \cdot \hat{r}) = g\Phi(\hat{r})$. In this case, the frame-averaging sum collapses such that

$$
\begin{aligned}
\langle \Phi \rangle_{\mathcal{F}}(\hat{r}) &= \frac{1}{|\mathcal{F}(\hat{r})|} \sum_{g \in \mathcal{F}(\hat{r})} g \cdot \Phi\left(g^{-1}\hat{r}\right) \\
&= \frac{1}{|\text{Stab}_{\text{SO}(3)}(\hat{v})|} \sum_{g \in \text{Stab}_{\text{SO}(3)}(\hat{v})} hg \cdot \Phi\left((hg)^{-1} \cdot \hat{r}\right) \\
&= \frac{1}{|\text{Stab}_{\text{SO}(3)}(\hat{v})|} \sum_{g \in \text{Stab}_{\text{SO}(3)}(\hat{v})} hg \cdot \Phi\left(g^{-1}h^{-1} \cdot \hat{r}\right) \\
&= \frac{1}{|\text{Stab}_{\text{SO}(3)}(\hat{v})|} \sum_{g \in \text{Stab}_{\text{SO}(3)}(\hat{v})} hgg^{-1} \cdot \Phi\left(h^{-1} \cdot \hat{r}\right) \\
&= \frac{1}{|\text{Stab}_{\text{SO}(3)}(\hat{v})|} \sum_{g \in \text{Stab}_{\text{SO}(3)}(\hat{v})} h \cdot \Phi\left(h^{-1} \cdot \hat{r}\right) \\
&= h \cdot \Phi\left(h^{-1} \cdot \hat{r}\right).
\end{aligned}
\tag{18}
$$

Thus, the entire frame reduces to the single rotation $h$ making the computation simple and tractable. We then define the $\mathcal{F}$ built upon the $\text{Stab}_{\text{SO}(3)}(r)$-equivariant $\Phi$ as the *SO(2) local frame*.

## D SO(2) OPERATIONS AND SO(3) OPERATIONS

### D.1 SO(2) LINEAR AND SO(3) TP WITH SPHERICAL HARMONICS

The relationship between SO(2) linear operations defined with a local frame and the SO(3) tensor product between irreducible representations and spherical harmonics has been investigated in the eSCN paper (34). For completeness, we briefly rehearse their proof here. If so, below we first provide the specific set of parameters for the SO(2) linear operation with SO(2) local frames that can exactly reproduce the results of the SO(3) TP with transformed CG coefficients. We will then explain why SO(2) linear operation with SO(2) local frames has the capacity to replicate it. Note that it is not to say that SO(2) linear with SO(2) local frame is definitely better, but it should have the potential to be at least the same. When performing SO(3) tensor product between the rotated SO(3) irreps and rotated spherical harmonics, the rotated spherical harmonics will be zero in the position with $m \neq 0$, and 1 for the position $m = 0$. Therefore, we can treat it as the corresponding transformed CG coefficients are nonzero for the following position.

$$
\left(\mathbf{c}_{l_i,l_f,l_o}\right)_m = \begin{cases} \mathbf{C}^{(l_o,m)}_{(l_i,m),(l_f,0)} & \text{if } m > 0 \\ \mathbf{C}^{(l_o,0)}_{(l_i,0),(l_f,0)} & \text{if } m = 0 \\ \mathbf{C}^{(l_o,-m)}_{(l_i,m),(l_f,0)} & \text{if } m < 0 \end{cases}
\tag{19}
$$

In this way, the SO(3) TP can be represented as

$$
\mathbf{f}^{(l_o)}_{m_o} = \sum_{m_i} \left(\mathbf{x}^{(l_i)}_s\right)_{m_i} \mathbf{C}^{(l_o,m_o)}_{(l_i,m_i),(l_f,0)} \mathbf{h}_{l_i,l_f,l_o},
\tag{20}
$$

where $\mathbf{f}$ is the output irreps and $\mathbf{h}$ is the coefficients for each path.

From the proposition in 3.1 from eSCN (34), we have

$$
\mathbf{C}^{(l_o,m)}_{(l_i,m),(l_f,0)} = \mathbf{C}^{(l_o,-m)}_{(l_i,-m),(l_f,0)}, \quad \mathbf{C}^{(l_o,m)}_{(l_i,-m),(l_f,0)} = -\mathbf{C}^{(l_o,-m)}_{(l_i,m),(l_f,0)},
$$

and $m_o = m_i$ or $m_o = -m_i$.

By using this proposition, the previous equation can be deducted into

$$
\mathbf{f}^{(l_o)}_{m_o} = \sum_{m_i = m_o, -m_o} \left(\mathbf{x}^{(l_i)}_s\right)_{m_i} \mathbf{C}^{(l_o,m_o)}_{(l_i,m_i),(l_f,0)} \mathbf{h}_{l_i,l_f,l_o},
\tag{21}
$$

.

And then, if we write $\mathbf{f}_{m_o}^{(l_o)}$ and $\mathbf{f}_{-m_o}^{(l_o)}$ together, it will become

$$
\begin{pmatrix} f_{m_o}^{(l_o)} \\ f_{-m_o}^{(l_o)} \end{pmatrix} = \begin{pmatrix} \tilde{\mathbf{h}}_{mo}^{(l',l)} & -\tilde{\mathbf{h}}_{-m_o}^{(l',l)} \\ \tilde{\mathbf{h}}_{-m_o}^{(l',l)} & \tilde{\mathbf{h}}_{m_o}^{(l',l)} \end{pmatrix} \cdot \begin{pmatrix} \left(\mathbf{x}_s^{(l_i)}\right)_{m_o} \\ \left(\mathbf{x}_s^{(l_i)}\right)_{-m_o} \end{pmatrix}, \tag{22}
$$

where $w_m^{(l_i,l_o)} = \sum_{l_f} h_{l_i,l_f,l_o} \cdot (c_{l_i,l_f,l_o})_m$ is the weight in the SO(2) linear case for $m_o \neq 0$.

Furthermore, since the parameters in SO(2) linear are learnable, the model may discover another set of parameterizations with lower training loss, potentially achieving better performance. Conceptually, SO(3) TP with transformed CG corresponds to a particular point in the parameter space of the SO(2) linear with SO(2) local frame. If the optimal parameters can be identified within this space, the model should perform at least as well as with SO(3) TP.

### D.2   SO(2) GATE AND SO(3) GATE

**SO(3) Gate.**   The SO(3) Gate takes the features with $l = 0$ as the input of an MLP, and then provides a gate value that is multiplied on each SO(3) feature with $l > 0$ as well as the new SO(3) features with $l = 0$. For convenience, we assume that the local frame, designed to rotate consistently with the input molecule, is currently aligned with the $y$-axis direction, so that the rotation procedure can be omitted.

**SO(2) Gate.**   Under this setup, the SO(2) Gate uses $m = 0$ features as the input of its MLP, including both $l = 0$ features and $m = 0, l \neq 0$ features. The MLP then produces a separate gate value for each SO(2) feature with different $m$, as well as new $m = 0$ features.

**Comparison.**   Because the MLP in the SO(2) Gate includes all the inputs of the SO(3) Gate (i.e., $l = 0$ components), it can replicate the same behavior as the SO(3) Gate. However, since the MLP contains nonlinear functions and can produce nonlinear results for the $m = 0, l \neq 0$ parts, the SO(3) Gate cannot approximate the SO(2) Gate. By including additional higher-order $m = 0$ inputs and using nonlinear functions to produce $m = 0$ outputs, the SO(2) Gate can therefore learn mappings that the SO(3) Gate cannot express. The details are demonstrated in Table 5.

Table 5: Comparison between SO(2) Gate and SO(3) Gate.

| Aspect | SO(2) Gate | SO(3) Gate |
|---|---|---|
| **MLP Input** | $m = 0$ features (includes $l = 0$ and $m = 0, l > 0$) | $l = 0$ features only |
| **Gate Applied To** | Each SO(2) feature with different $m > 0$ | Each SO(3) feature with different $l > 0$ |
| **New Output Components** | $m = 0$ (including both $l = 0$ and $l > 0$) | $l = 0$ only |

## E   EQUIVARIANCE OF THE IRREP MAPPING

For the rotation within the local frame, an SO(2) rotation with angle $\alpha$ applied to features of order $m$ corresponds to the following rotation matrix:

$$
R_m(\alpha) = \begin{pmatrix} \cos(m\alpha) & -\sin(m\alpha) \\ \sin(m\alpha) & \cos(m\alpha) \end{pmatrix}. \tag{23}
$$

For SO(3) rotations acting on SO(3) irreps, if the rotation is restricted to the $z$-axis (i.e., the exact $m = 0$ feature for $l = 1$), it corresponds to the ZYZ Euler rotation with parameters $(\alpha, \beta, \gamma) = (\alpha, 0, 0)$. The corresponding rotation matrix is

$$
D_{mn}^{(\ell)}(\alpha, 0, 0) = e^{im\alpha}\delta_{mn}, \tag{24}
$$

for SO(3) irreps with angular momentum $\ell$.

With the change of basis $U_\ell$ derived in Equation (25) of the eSCN, the rotation matrix can be expressed in the same block-diagonal format as

$$D_{\text{real}}^{(\ell)}(\alpha) \;=\; U_\ell\, D_{\text{complex}}^{(\ell)}(\alpha, 0, 0)\, U_\ell^\top \;=\; \text{diag}(1, R_1(\alpha), R_2(\alpha), \ldots, R_\ell(\alpha)), \qquad (25)$$

where each block $R_m(\alpha)$ matches the SO(2) rotation matrix defined above.

Then, for the equivariance of the mapping operation $FR(\hat{r})$, the formal definition of equivariance for a map $f$ is

$$f(g \cdot x) \;=\; \rho(g)\, f(x), \qquad (26)$$

where $g \in$ SO(3) acts on the input, and $\rho(g)$ is the induced SO(2) action on the output irreps.

In our setting, $f$ denotes the operation mapping from SO(3) to SO(2) with the local frame defined as $FR(r_{ij})$. When $g \in$ SO(3) is applied to the input space, the corresponding stabilizer rotation is given by $g'g^{-1}$, where $g'$ is a rotation around the $z$-axis that maps the rotated direction vector $gr_{ij}$ back to the $z$-axis. Applying $g'g^{-1}$ to the rotated input $gx$ is equivalent to applying $g'$ to the original SO(3) irreps $x$ before rotation. As shown earlier, the rotation matrix for SO(3) features with only $\alpha \neq 0$ reduces to the SO(2) rotation form. Hence, the stabilizer rotation $g'$ serves as the corresponding $\rho(g)$ acting on the output SO(2) irreps.

## F  HYPERPARAMETER SETTINGS AND MODEL ARCHITECTURES

### F.1  QH9

In the QH9-stable-id setting, the training, validation and test sets are randomly split. In contrast, the QH9-stable-ood settings use a split based on molecular size, with validation and test sets consisting of molecules larger than those in the training set. The QH9-dynamic-300k dataset is an extended version of the QH9-dynamic benchmark, comprising 300k molecular geometries generated from 3k molecules, each with a molecular dynamics trajectory of 100 steps. In the QH9-dynamic-300k-geo setting, the training, validation, and test sets consist of the same molecules but different geometries sampled from their respective trajectories. In contrast, the QH9-dynamic-300k-mol setting is split based on different molecules, with all geometries from a given trajectory assigned to the same split. This setup is designed to evaluate the model's ability to generalize to unseen molecules with diverse geometries. We follow the model and training hyperparameters in Table 6 to perform our experiments.

Table 6: Hyperparameters in QH9.

| Hyperparameters | Values/search space |
|---|---|
| Batch size | 32 |
| Cutoff distance (Bohr) | 15 |
| Initial learning rate | 5e-4 |
| Final learning rate | 1e-7 |
| Learning rate strategy | linear schedule with warmup |
| Learning rate warmup batches | 1,000 |
| # of batches in training | 26,000 |
| # of layers | 3 |
| $L_{max}$ | 4 |
| hidden irreps | 256x0e+128x1e+64x2e+32x3e+16x4e |
| hidden mirreps | 1024x0m+256x1m+64x2m+32x3m+16x4m |
| hidden ffn mirreps | 2048x0m+512x1m+256x2m+64x3m+32x4m |

## F.2 MD17

The statistics of MD17 is shown in Table 7. We follow the model and training hyperparameters in Table 8 to perform our experiments. Specifically, for the model molecule, we choose a smaller model since the number of training geometries is smaller. We choose hidden irreps to be 64x0e+32x1e+16x2e+8x3e+8x4e, hidden mirreps to be 64x0m+32x1m+16x2m+8x3m+8x4m, and hidden ffn mirreps to be 64x0m+32x1m+16x2m+8x3m+8x4m.

Table 7: The statistics of MD17 dataset.

| Dataset | Molecule | # of structures | Train | Val | Test |
|---------|----------|-----------------|-------|-----|------|
| Water | $H_2O$ | 4,900 | 500 | 500 | 3,900 |
| Ethanol | $C_2H_5OH$ | 30,000 | 25,000 | 500 | 4,500 |
| Malondialdehyde | $CH_2(CHO)_2$ | 26,978 | 25,000 | 500 | 1,478 |
| Uracil | $C_4H_4N_2O_2$ | 30,000 | 25,000 | 500 | 4,500 |

Table 8: Hyperparameters in MD17.

| Hyperparameters | Values/search space |
|-----------------|---------------------|
| Batch size | 5, 10, 16 |
| Cutoff distance (Bohr) | 15 |
| Initial learning rate | 1e-3, 5e-4 |
| Final learning rate | 1e-6, 1e-7 |
| Learning rate strategy | linear schedule with warmup |
| Learning rate warmup batches | 1,000 |
| # of batches in training | 20,000 |
| # of layers | 2, 3 |
| $L_{max}$ | 4 |
| hidden irreps | 256x0e+128x1e+64x2e+32x3e+16x4e |
| hidden mirreps | 1024x0m+256x1m+64x2m+32x3m+16x4m |
| hidden ffn mirreps | 2048x0m+512x1m+256x2m+64x3m+32x4m |

