# OpenReview forum: "Efficient Prediction of SO(3)-Equivariant Hamiltonian Matrices via SO(2) Local Frames"
_ICLR.cc/2026/Conference — Submitted to ICLR 2026_

### Official Review · Reviewer_cJL7 · 2025-10-28

**Soundness:** 2
**Presentation:** 2
**Contribution:** 2
**Rating:** 2
**Confidence:** 5

**Summary:**

This paper introduces QHNetV2, a neural network designed for predicting Hamiltonian matrices. QHNetV2 achieves global SO(3) equivariance through a set of efficient SO(2)-equivariant operations.

**Strengths:**

* The paper is well-structured and clearly written, offering an accessible and concise overview of both the problem and the proposed solution.
* The results presented on the two datasets show that the model performs well.

**Weaknesses:**

* The main claimed innovation of this paper lies in using SO(2) to efficiently predict Hamiltonians while preserving equivariance. However, techniques involving SO(2) tensor-product equivariance have already been extensively studied in both machine learning force field and Hamiltonian prediction research. Therefore, the contribution of this work appears rather trivial and not significant. What is the specific difference between the SO(2)-based module design in this paper and that in [1-3]?
* The experiments were conducted on only two datasets. It would strengthen the paper to include additional evaluations on other datasets, such as $\nabla^2$ DFT [4], to further demonstrate the model’s generality.
* The PhiSNet results reported for MD17 are not the original ones, but rather the reproduced results from the QHNet implementation. Such a comparison is unfair, and I could not find any statement in the paper clarifying that the PhiSNet results were extracted from QHNet paper.

[1] Passaro S, Zitnick C L. Reducing SO (3) convolutions to SO (2) for efficient equivariant GNNs[C]//International conference on machine learning. PMLR, 2023: 27420-27438.

[2] Liao Y L, Wood B, Das A, et al. Equiformerv2: Improved equivariant transformer for scaling to higher-degree representations[J]. arXiv preprint arXiv:2306.12059, 2023.

[3] Wang Y, Li H, Tang Z, et al. Deeph-2: Enhancing deep-learning electronic structure via an equivariant local-coordinate transformer[J]. arXiv preprint arXiv:2401.17015, 2024.

[4] Khrabrov K, Ber A, Tsypin A, et al. $\nabla^ 2$ DFT: A Universal Quantum Chemistry Dataset of Drug-Like Molecules and a Benchmark for Neural Network Potentials[J]. Advances in Neural Information Processing Systems, 2024, 37: 36869-36889.

**Questions:**

See Weaknesses.

---

> ### Author Response · Authors · 2025-11-28
> **Reply to reviewer cJL7**
>
> Thanks for your comments, and we provide our reply here hoping it can resolve your concerns.
>
> W1. We respectfully disagree with the comments. To improve computational efficiency, ESCN proposes to reduce the SO(3) tensor product into SO(2) in this case that the SO(3) tensor product can be equivalently replaced by a SO(2) Linear.  Previous works like eSCN, EquiformerV2, eSEN and DeepH-2, they develop their model based on this mechanism to enable efficient model design. Motivated by the efficiency gains achieved through SO(2)-based operations, in this work we further generalize the SO(2) operations beyond SO(2) linear layers like **SO(2) TP which has not been introduced in related works before**. We show that any SO(2) operations can be incorporated within these SO(2) local frames while maintaining global SO(3) equivariance, enabled by a minimal frame-averaging mechanism. In addition, these SO(2) local frames can be constructed not only on edges but also on nodes or other locations where a well-defined directional reference can be established.
>
> W2. Thanks for the suggestion. Due to the limited time of rebuttal, we add the reference and will include the experiments on $\nabla^2$DFT in the future. Thanks for your understanding.
>
> W3. Thanks for raising the concerns. We wish to clarify that we follow the experimental settings of QHNet, and the experimental results of QHNet and PhiSNet are from Table 1 in QHNet experiments. Since increasing the number of iterations can improve the final performance, to make a fair comparison, the total number of iterations are fixed for these experiments with a total 200,000 iterations. The original implementation and experiments of PhiSNet follows the settings of using reduce on the Plateau and won't stop until the learning rate is reduced to or achieving a total number of iterations. This will lead to a very large number of iterations for training, and the number of iterations are not fixed and pretty large for all the models. Therefore, we follow the experimental setting of QHNet to conduct the experiments. We will emphasize this in our manuscript to avoid any confusion. Hope you can understand it. We list the results of different training sets here. You will find that increasing the total number of batches will increase the performance based on the experiments of QHNet. For QHNetV2, although the total number of iterations is 200,000 which is smaller than 1,000,000 for QHNet and RLROP for PhiSNet (surely longer than 200,000 iterations) , the performance is at a comparable level for Ethanol and Malondialdehyde.
>
> | Dataset   | Method    | Training Strategies   | H [10⁻⁶ Eₕ] ↓ | ε [10⁻⁶ Eₕ] ↓ | ψ [10⁻²] ↑ |
> |---|----|-----|-----|------|----|
> | Water     | QHNet*    | RLROP     | 10.36  | 36.21   | 99.99  |
> |   | PhiSNet   | RLROP  | 17.59 | 85.53    | -    |
> |   | PhiSNet   | LSW (10,000, 200,000)   | 17.59   | 85.53   | 100.00     |
> |   | QHNet  | LSW (10,000, 200,000)  | 10.36    | 36.21         | 99.99      |
> |   | SPHNet    | LSW (10,000, 200,000)   | 23.18         | 182.29        | 100.00     |
> |   | QHNetV2   | LSW (10,000, 200,200)    | 22.55    | 106.64        | 99.99      |
> | Ethanol   | QHNet     | RLROP     | 13.12    | 51.80    | 99.99      |
> |     | PhiSNet   | RLROP     | 12.15  | 62.75         | -      |
> |    | PhiSNet   | LSW (10,000, 200,000)     | 20.09         | 102.04        | 99.81      |
> |   | QHNet     | LSW (10,000, 200,000)      | 20.91         | 81.03         | 99.99      |
> |   | SPHNet    | LSW (10,000, 200,000)     | 20.27         | 97.03         | 99.99      |
> |  | QHNetV2   | LSW (10,000, 200,000)     | 12.05         | 70.46         | 99.99      |
> |   | QHNet     | LSW (10,000, 1,000,000)    | 12.78         | 62.97         | 99.99      |
> | Malondialdehyde    | QHNet     | RLROP       | 13.18         | 51.54         | 99.95      |
> |   | PhiSNet   | RLROP      | 12.32    | 73.50         | -    |
> | | PhiSNet   | LSW (10,000, 200,000)           | 21.31         | 100.60        | 100.00     |
> |   | QHNet     | LSW (10,000, 200,000)           | 21.52         | 95.20         | 99.98      |
> |  | SPHNet    | LSW (10,000, 200,000)           | 20.67         | 95.77         | 99.99      |
> |   | QHNetV2   | LSW (10,000, 200,000)           | 12.20         | 62.46         | 99.92      |
> |  | QHNet     | LSW (10,000, 1,000,000)         | 11.97         | 55.57         | 99.94      |
> | Uracil             | PhiSNet   | RLROP                           | 10.73         | 84.03         | -    |
> |    | PhiSNet   | LSW (10,000, 200,000)           | 18.65         | 143.36        | 99.96      |
> |    | QHNet     | LSW (10,000, 200,000)           | 11.35         | 113.44        | 99.99      |
> |    | SPHNet    | LSW (10,000, 200,000)           | 19.36         | 118.21        | 100.00     |
> |    | QHNetV2   | LSW (10,000, 200,000)           | 10.38         | 107.42        | 99.91      |
> |    | QHNet     | LSW (10,000, 1,000,000)         | 9.96          | 66.75         | 99.95      |

---

### Official Review · Reviewer_gAzZ · 2025-10-30

**Soundness:** 3
**Presentation:** 3
**Contribution:** 2
**Rating:** 6
**Confidence:** 4

**Summary:**

The paper proposes QHNetV2, a Hamiltonian prediction network that achieves global SO(3) equivariance via locally constructed SO(2) frames. By replacing the heavy Clebsch–Gordan tensor products with efficient SO(2)-equivariant operations (Linear, Gate, LayerNorm, Tensor Product), the model significantly reduces computational complexity while maintaining physical symmetry. Experiments on QH9 and MD17 show competitive accuracy.

**Strengths:**

1. Use of local SO(2) frames to achieve global SO(3) equivariance to reduce costs
2. Improved efficiency gains with comparable or better accuracy.
3. Promising direction for scalable learning of quantum Hamiltonians.

**Weaknesses:**

1. While the idea of achieving global SO(3) equivariance through SO(2) operations is elegant, it is not entirely novel. Similar concepts have been explored in previous works, such as eSCN, which the authors themselves acknowledge. The main contribution here lies in applying this framework to Hamiltonian prediction. As a result, the methodological innovation is somewhat limited, and the paper would benefit from a clearer articulation of what new theoretical or algorithmic insights distinguish it from existing SO(2)-based approaches.
2. In principle, replacing SO(3) tensor products with SO(2) operations should substantially reduce computational cost. However, the manuscript provides only limited benchmarking evidence, and the reported efficiency gains (e.g., 4.34× in Table 3) appear modest given the expected complexity reduction. It would be valuable to see a more systematic analysis of scalability — for instance, how efficiency varies with model size or with increasing molecular size and number of atoms (e.g., on MD17 systems).
3. The experimental setup appears somewhat narrow. The QM9 dataset primarily contains small molecules with similar compositions and sizes, which may make generalization within this dataset relatively easy. The MD17 trajectories also focus on closely related structures, which test temporal consistency rather than true out-of-distribution generalization. It would greatly strengthen the paper to evaluate the trained models on unseen molecular systems — for example, testing a model trained on QM9 to predict Hamiltonians of larger alkanes (CH4, C2H6, …, C20H42, …). Such experiments would help demonstrate whether the model captures transferable physical relationships rather than dataset-specific correlations.

**Questions:**

My questions are listed in the weakness section, here I only have one more question:
1. How do you think about the difficult to generalize this model to materials that have periodic boundary condition? Is there any technical issue you have to solve before you can do it?

---

> ### Author Response · Authors · 2025-11-28
> **Reply to reviewer gAzZ**
>
> W1. Thanks for the comments, and we simply summarize our contributions about SO(2) operations here.  As stated in the manuscripts, eSCN first uses the SO(2) Linear on edges with the motivation to replace the SO(3) TP to accelerate the model. Motivated by the efficiency gains achieved through SO(2)-based operations, in this work we further generalize the SO(2) operations beyond SO(2)-linear layers. We show that any SO(2) operations can be incorporated within these SO(2) local frames while maintaining global SO(3) equivariance, enabled by a minimal frame-averaging mechanism. In addition, these SO(2) local frames can be constructed not only on edges but also on nodes or other locations where a well-defined directional reference can be established.
>
>
> W2. Thanks for your valuable comments on the efficiency study. We provide more concrete results in the table below. Please note that these experiments were conducted on an RTX 6000, which is slightly faster than the A6000 GPU used in the manuscript. Nevertheless, we observe a relatively consistent inference-time ratio across different molecule sizes between these models if . We would like to emphasize that a 4× speedup is substantial, especially considering that many other components in the network—such as linear layers on edges and nodes—also contribute to the total computation cost beyond the SO(3) tensor products and SO(2) operations.
>
>
> | Atoms | Model   | Batch Size | Speed (Samples/s) | Memory (GB/sample) | Speedup |
> |-------|---------|-------------|--------------------|---------------------|---------|
> | 9     | QHNet   | 608         | 227.09             | 41.0351             | 1.00x   |
> | 9     | QHNetV2 | 2688        | 1053.53            | 9.2796              | 4.64x   |
> | 9     | SPHNet  | 1344        | 851.62             | 23.8666             | 3.75x   |
> | 14    | QHNet   | 192         | 93.47              | 103.7581            | 1.00x   |
> | 14    | QHNetV2 | 1152        | 426.13             | 23.2532             | 4.56x   |
> | 14    | SPHNet  | 480         | 371.41             | 55.7157             | 3.97x   |
> | 19    | QHNet   | 64          | 50.93              | 200.0575            | 1.00x   |
> | 19    | QHNetV2 | 544         | 233.55             | 43.6480             | 4.59x   |
> | 19    | SPHNet  | 224         | 206.36             | 100.5308            | 4.05x   |
> | 24    | QHNet   | 64          | 31.76              | 312.6754            | 1.00x   |
> | 24    | QHNetV2 | 352         | 146.09             | 70.2527             | 4.60x   |
> | 24    | SPHNet  | 160         | 131.00             | 155.6871            | 4.13x   |
> | 29    | QHNet   | 64          | 21.71              | 445.5830            | 1.00x   |
> | 29    | QHNetV2 | 224         | 97.53              | 103.3498            | 4.49x   |
> | 29    | SPHNet  | 96          | 91.00              | 224.0591            | 4.19x   |
>
>
>
> W3. Thanks for pointing out the experimental case of OOD generalization. And we also think it is a vital problem. To evaluate the OOD settings, the QH9-stable-ood provides an experimental case for this important scenario. The split of QH9-stable-ood is shown in QH9, the training sets only contain molecules with less than 20 atoms, while the larger molecules range from 21 atoms to 31 atoms are provided in the validation and test sets. And you will find that in Table 1, the corresponding performance on the Hamiltonian and eigen energy demonstrate valid transferability results. Note that the molecules in the in-distribution (ID) and out-of-distribution (OOD) test sets are not the same—the ID split contains smaller molecules in their test sets. Therefore, the numerical results from the ID test set and those from the OOD test set are not directly comparable.
>
> Q1. Thanks for asking this question. The key will depend on how to define the SO(2) local frame for the input geometric graph with periodicity. If you consider building the input graph with a cutoff to form a multi-edge graph such as CGCNN [1], it needs to take care about the multi-edge graph where the multi-edge between the nodes will have different directions denoting the connections between atoms within different cells. If you further consider the periodicity of the materials using the lattice vectors such as MatFormer[2], the model may need to be modified to encode these lattice vectors before applying this model on materials. If the periodicity constraint is not considered, directly extending the neighboring-cell atoms as additional nodes to construct the cutoff graph is also a valid approach. In this case, no further modification is required. Thanks!
>
> [1] Crystal Graph Convolutional Neural Networks for an Accurate and Interpretable Prediction of Material Properties.
> [2] Periodic Graph Transformers for Crystal Material Property Prediction

---

### Official Review · Reviewer_mpT8 · 2025-10-31

**Soundness:** 3
**Presentation:** 3
**Contribution:** 3
**Rating:** 6
**Confidence:** 4

**Summary:**

This paper proposes QHNetV2, an efficient equivariant neural network for predicting SO(3)-equivariant Hamiltonian matrices. The method replaces all expensive SO(3) tensor product operations with SO(2) operations performed in local reference frames. Each node, edge, and node pair defines its own local SO(2) frame through minimal-frame canonicalization and frame averaging. Within each frame, all message passing, linear, gate, and feed-forward (FFN) operations are executed in SO(2)-equivariant form, fully removing the need for SO(3) Clebsch–Gordan tensor products (while still using efficient SO(2) tensor products for feature mixing). Experiments on the QH9 and MD17 molecular datasets demonstrate strong predictive accuracy and a significant speedup over QHNet.

**Strengths:**

1. Comprehensive use of SO(2) operations. The model systematically generalizes SO(2)-equivariant operations beyond the message passing step, extending them to the FFN, gate, and linear layers, and allowing an entirely SO(2)-based neural architecture. This contributes to higher efficiency and stability.

2. Strong and broad empirical performance. The model achieves state-of-the-art accuracy on QH9 and competitive results on MD17, showing that the proposed SO(2) local frame formulation generalizes well across different molecular prediction tasks. Quantitative efficiency results indicate a significant speedup over QHNet, likely attributable to the elimination of SO(3) tensor products.

3. Solid engineering execution. The design of SO(2) linear, gate, and FFN modules is unified and modular. The method is cleanly implemented and could be easily generalized to other physics-informed learning tasks.

**Weaknesses:**

1. Limited novelty of the core operator. The SO(2) linear/tensor-product mechanism has already been introduced and optimized in eSCN [1] and subsequently adopted in EquiformerV2 [2] and eSEN [3]; moreover, it has been used in downstream Hamiltonian modeling such as DeepH-2 (which employs EquiformerV2’s SO(2) convolution) [4]. As a result, the present paper’s contribution appears primarily as extending these existing SO(2) operations to additional layers (FFN, gate) and applying them to Hamiltonian prediction. The authors should clarify what is theoretically or representationally new beyond these prior SO(2)-equivariant frameworks and implementations (including DeepH-2).

2. No dynamic weighting (attention) mechanism. Equivariant Transformers such as EquiformerV2 and eSEN show that dynamic weighting (attention) improves adaptivity and expressivity. QHNetV2 instead uses a deterministic (non-attention-based) aggregation scheme, without dynamic weighting. Why not adopt dynamic weights? Is it for computational reasons or due to theoretical constraints of SO(2) operations?

3. Potential loss of inversion equivariance. The Hamiltonian requires strict inversion symmetry (even/odd parity). SO(2) operations handle rotations but not reflections. How does the model preserve inversion equivariance? Moreover, since the features are all even functions, how can the network capture odd (antisymmetric) matrix components?

4. Experiments are only on molecular datasets (QH9, MD17). Can the proposed SO(2) local frame generalize to periodic or crystalline systems, where translational and lattice symmetries play a role?

5. Realistic Hamiltonians often include spin–orbit coupling, which requires complex-valued spinor representations (SU(2) symmetry). Can this SO(2)-based model handle such cases, or would an SU(2)-equivariant extension be necessary?

6. Possible geometric degeneracy in local frame construction.
The paper adopts a “minimal frame averaging” approach to define local SO(2) reference frames from aggregated neighborhood geometry. However, when atomic neighborhoods are nearly isotropic (e.g., central atoms in symmetric environments), the averaged direction may vanish or fluctuate, leading to unstable or ill-defined local frames. Moreover, such averaging can smooth out fine local structural details and erase genuine geometric anisotropy, causing the constructed frame to lose its physical interpretability. This degeneracy risks breaking rotational consistency and numerically violating SO(3) equivariance. The authors should clarify how such issues are mitigated.

[1] Passaro, Saro, and C. Lawrence Zitnick. "Reducing SO(3) convolutions to SO(2) for efficient equivariant GNNs." ICML 2023.
[2] Liao, Y. L., Wood, B., Das, A., & Smidt, T. (2023). Equiformerv2: Improved equivariant transformer for scaling to higher-degree representations. ICLR 2024.
[3] Fu, X., Wood, B. M., Barroso-Luque, L., Levine, D. S., Gao, M., Dzamba, M., & Zitnick, C. L. (2025). Learning smooth and expressive interatomic potentials for physical property prediction. ICML 2025.
[4] Wang, Y., Li, Y., Tang, Z., Li, H., Yuan, Z., Tao, H., ... & Xu, Y. (2024). Universal materials model of deep-learning density functional theory Hamiltonian. Science Bulletin, 69(16), 2514–2521.

**Questions:**

Could you please clarify where the Hamiltonian labels for the MD17 dataset come from? The original dataset does not include such labels, and the paper does not specify whether they were generated by the authors themselves.

---

> ### Author Response · Authors · 2025-11-28
> **Reply to reviewer mpT8 (1/2)**
>
> Thanks for your valuable comments. If we understand it correctly, W1 is about the contributions of SO(2) frames. W2, W3, W4, and W5 are about potential enhancements for more powerful models and general cases. We reply to them point to point here. And we will include a paragraph about further direction for these potential enhancements. Thanks!
>
> W1.  Thanks for raising this concern about the usage of SO(2) local frame technique used in this model and existing models. To improve computational efficiency, ESCN proposes to reduce the SO(3) tensor product into SO(2) in this case that the SO(3) tensor product can be equivalently replaced by a SO(2) Linear.  Previous works like eSCN, EquiformerV2, eSEN and DeepH-2, they develop their model based on this mechanism to enable efficient model design. Motivated by the efficiency gains achieved through SO(2)-based operations, in this work we further generalize the SO(2) operations beyond SO(2) linear layers. We show that any SO(2) operations can be incorporated within these SO(2) local frames while maintaining global SO(3) equivariance, enabled by a minimal frame-averaging mechanism. In addition, these SO(2) local frames can be constructed not only on edges but also on nodes or other locations where a well-defined directional reference can be established.
>
>
> W2. Thanks for the comment. Dynamic weighting mechanisms—such as the attention module used in EquiformerV2—are indeed effective for improving model performance with reasonable computational cost. We will consider incorporating such mechanisms as a potential direction for future enhancement of our network. Similar models, including eSCN and eSEN, may also benefit from integrating attention. Thank you for the valuable suggestions.
>
> W3. Thanks for your comments, and it is an interesting potential enhancement design. In the current Hamiltonian datasets, MD17 and QH9, the Hamiltonian matrices can be treated as SE(3)-equivariant which do not exhibit inversion symmetry. Based on this, we build our SE(3) equivariant network without incorporating inversion symmetry for the problem.
> But for sure, this task can also be treated as O(3) symmetry.
> If so, the corresponding SO(2) irreps needed to be designed with parity symmetry.
> That's to say, for the SO(3) irreps, it will have the parity symmetry, and when performing the transformation from SO(3) irreps to SO(2) irreps based on the rotation with the wigner D matrix, the parity needed to be considered.  And then, the SO(2) linear can only be applied on the SO(2) irreps with the same parity and order.  This part may need to be further investigated in the future to see whether such a change in the model design will bring performance improvement or not. Currently, the parity symmetry has not been introduced for models with SO(2) yet.
>
> W4. Thanks for your question. For sure, such operations can definitely be applied for periodic or crystalline systems. The key will depend on how to define the SO(2) local frame for the input geometric graph with periodicity. If you consider building the input graph with a cutoff to form a multi-edge graph such as CGCNN [1], it needs to take care about the multi-edge graph where the multi-edge between the nodes will have different directions denoting the connections between atoms within different cells. If you further consider the periodicity of the materials using the lattice vectors such as MatFormer[2], the model may need to be modified to encode these lattice vectors before applying this model on materials. If the periodicity constraint is not considered, directly extending the neighboring-cell atoms as additional nodes to construct the cutoff graph is also a valid approach. In this case, no further modification is required. Thank you!
>
> W5.  Thanks for your comments. The spin–orbital coupling term constitutes an additional Hamiltonian component that becomes essential when analyzing magnetic materials. Addressing this interaction requires incorporating SU(2) symmetry, as spin degrees of freedom transform under the SU(2) group. To extend our framework to this setting, the final tensor expansion layer would likely need to be generalized or replaced with a more expressive expansion technique capable of handling SU(2)-equivariant features such as Wigner-Eckart layer in DeepH-E3, while the previous model architecture can be kept the same. Developing such an SU(2)-aware tensor expansion would be an important direction for future work.

---

> > ### Author Response · Authors · 2025-11-28
> > **Reply to reviewer mpT8 (2/2)**
> >
> > W6. This is a very important comment. Highly symmetric configurations are among the most challenging cases for frame-construction techniques, and many existing methods can fail under such symmetry. For example, SVD-based frame construction often becomes ill-posed because the singular vectors cannot uniquely determine a coordinate system in these degenerate situations. In our work, we address this issue by performing SO(2) operations on all neighboring atoms and then averaging the results. This approach effectively aggregates multiple candidate frames, allowing us to resolve the symmetry-induced ambiguity. The computational complexity of this procedure is $O(N)$ per node, where $N$ is the number of atoms. For practical applications, we simplify this process by selecting the nearest neighbor to construct the frame, which provides a significantly faster yet still effective approximation.
> >
> >
> > Q1. Thanks for raising this question. The label of Hamiltonian matrix for MD17 is from SchNorb, which is the first paper in this direction, it proposes an invariant model with data augmentation for the hamitonian matrix. The generated dataset provides a valuable dataset for the following studies of the equivariant model on predicting such complex important equivariant matrix properties.
> >
> > [1] Crystal Graph Convolutional Neural Networks for an Accurate and Interpretable Prediction of Material Properties.
> > [2] Periodic Graph Transformers for Crystal Material Property Prediction
> > [3] General framework for E(3)-equivariant neural network representation of density functional theory Hamiltonian

---

### Official Review · Reviewer_b21d · 2025-10-31

**Soundness:** 2
**Presentation:** 2
**Contribution:** 2
**Rating:** 4
**Confidence:** 4

**Summary:**

The paper targets fast, accurate prediction of Kohn–Sham Hamiltonian matrices. It introduces SO(2) local frames that allow most computations to be carried out with SO(2)-equivariant layers while preserving global SO(3) equivariance via canonicalization/minimal frame averaging. The approach avoids explicit SO(3) CG tensor products in message passing and feature updates and adds several SO(2)-equivariant building blocks—Linear, Gate, LayerNorm, and a (continuous) SO(2) tensor product (TP) for feature fusion. Architecture-wise (Fig. 2, p. 6), node features are updated in the global space with node‑wise interactions + SO(2) TP in a local node frame, whereas off‑diagonal (pair) features—which dominate the Hamiltonian—are updated and kept in their SO(2) pair frames using an SO(2) FFN. Experiments on QH9 and MD17 show  lower Hamiltonian‑MAE vs prior art and speedups over SO(3)-TP methods (Tables 1–3, pp. 9, 14)

**Strengths:**

1. The theory of connecting the SO(2) to minimal frame averaging is interesting. Appendix C shows how frame averaging collapses to the canonical rotation when the local model is equivariant to the stabilizer.
2. Empirical improvements are clear and practically relevant. Laudable MAE reductions on QH9 (including OOD) and solid speedups vs a strong SO(3)-TP baseline demonstrate usefulness (Table 1 & 3, pp. 9, 14).

**Weaknesses:**

1. The scalability of the model remains unclear. The authors should evaluate and compare performance with SPHNet and QHNet across systems of increasing Hamiltonian size to assess how well the proposed approach scales.
2. The discussion of related work is underdeveloped. Given that the contribution lies within the design of equivariant architectures, the paper should discuss prior SO(3) and SO(2)-equivariant models and spherical scalarization models (e.g., e3nn, SE(3)-Transformers, TFN, Allegro, SEGNN, GotenNet, ViSNet) to contextualize how the proposed approach differs in representation efficiency, inductive bias, and computational scaling.
3. Ablation is insufficient. The ablation analysis in its current form is too coarse to justify the key architectural claims. It toggles components like the SO(2) TP and SO(2) FFN but never clarifies what replaces them when removed—whether the block is skipped, replaced with an SO(2) Linear, or swapped with an SO(3) TP—which makes the results in Table 4 uninterpretable. The ablation also conflates multiple design factors: it does not isolate the effect of the continuous SO(2) TP’s parameters (v, M), the “keep-in-frame” vs. “re-project” strategy for pair features, or different node-frame constructions (nearest-neighbor vs. averaged). Moreover no runtime or memory profile connects the claimed efficiency gains to actual costs. A more systematic ablation—explicit replacements, controlled cost–benefit sweeps, and frame-robustness tests—would make the paper’s design rationale far more convincing.
Minor:
1. Table 3 (p. 14) uses “maximum available batch size” on a single A6000. Because max batch differs by model/memory footprint, throughput (samples/s) could be confounded. A stronger comparison would fix (i) hardware/precision, (ii) batch size, (iii) sequence length/graph size bins, and report per‑sample latency and FLOPs to isolate algorithmic speedups. Please also report precision (FP32/FP16/BF16) and cudnn/e3nn kernel variants used

**Questions:**

1. The node frame is defined by the nearest-neighbor direction (Eq. 13, p. 7), which the paper acknowledges may introduce discontinuities when neighbor assignments change. While future remedies such as averaging over O(n) frames are mentioned, the current formulation may still experience gauge flips. Could you quantify how frequently these occur in practice and describe their effect on training stability and convergence?
2. The authors of eSCN derive the SO(2) convolution complexity as $O(L^3)$. Could you similarly derive and report the computational complexity of your proposed SO(2) tensor product (TP) in terms of $L$? (i.e., for a given $L$, first apply a rotation to align features into the SO(2) subspace, perform the SO(2) tensor product there, then rotate back to the original frame—please include these rotation steps in the overall computational complexity analysis) In practice, what value of $v$ is used in the many-body expansion? How do runtime and memory scale with $M_{\max}$, $v$, and the number of channels? The ablation in Table 4 (p. 14) is unclear—what exactly does “no SO(2) TP” mean? Was it replaced with an SO(3) TP or omitted entirely?
3.Can the expansion module also leverage SO(2) representations to further reduce computational cost?
4. Could the authors provide a detailed runtime and memory profile of the network? Specifically, which modules (e.g., SO(2) TP, FFN, gating, or LayerNorm blocks) dominate the computation or memory usage? Such profiling would be valuable for identifying future optimization and scaling directions.

---

> ### Author Response · Authors · 2025-11-28
> **Reply to reviewer b21d (1/3)**
>
> Thank you for your valuable comments. We provide our detailed rebuttal below, and we appreciate your patience throughout this process.
> W1 & W4. Thanks for your suggestions. If we understand correctly, the scalability of the model primarily refers to its computational efficiency or inference time as the Hamiltonian size or molecule size increases. The batch size is fixed into 32, the precision is FP32, e3nn version is 0.5.6 and cudnn version is 90100. Here we report the inference time for each batch. The measurement includes only the model’s forward pass, with all data-loading overhead removed. We observe that QHNetV2 achieves slightly higher FLOPS than SPHNet, while also providing slightly faster inference since it does not rely on TP or sparse TP operations. As the number of atoms increases, the inference-time ratio is stable.
> | Number of atoms within the molecule | QHNetV2 inference time | QHNetV2 FLOPS | SPHNet inference time | SPHNet FLOPS | Time Ratio (QHNetV2 / SPHNet) |
> |------------------------------------|-------------------------|---------------|------------------------|---------------|-------------------------------|
> | 9                                  | 157.44 ms               | 1.39G         | 173.10 ms              | 1.47G         | 0.909                         |
> | 14                                 | 281.36 ms               | 3.39G         | 338.65 ms              | 3.32G         | 0.831                         |
> | 19                                 | 467.85 ms               | 6.27G         | 568.56 ms              | 5.91G         | 0.823                         |
> | 24                                 | 694.32 ms               | 10.01G        | 871.98 ms              | 9.21G         | 0.796                         |
> | 29                                 | 979.09 ms               | 14.63G        | 1245.04 ms             | 13.23G        | 0.786                         |
>
>
> W2: Thank you for the suggestion to include additional related works on equivariant networks. We agree that this is an important aspect, and we have incorporated these references into Section 4 of the revised manuscript. For convenience, we also list them below.
>
>
> Since intrinsic symmetries present in physical systems play an important role in modeling, equivariant neural networks are explicitly designed to encode these symmetries directly into the architecture. By construction, these models preserve equivariant features throughout all layers, ensuring that symmetry-consistent representations are maintained.
> For Cartesian equivariant models such as PaiNN, ViSNet, and GotenNet, the architectures rely on scalars and vector features within the model. These models are both efficient and powerful, enabling the models to achieve strong performance across a wide range of tasks.
> In contrast, spherical equivariant models are built on SO(3) irreducible representations and make extensive use of spherical harmonics and tensor products to rigorously encode rotational symmetries. Examples include TFN, SEGNN, SE(3)-Transformers, NequIP, MACE, Allegro, Equiformer. These models build their architectures around tensor products as the central mechanism for encoding directional information within the irreducible representations of the feature space. Although tensor products offer a powerful mechanism for encoding geometric information, their computational cost has been widely discussed as a major bottleneck with a computational complexity $O(L^6)$, where $L$ is the maximum degree of the input irreducible representations. To improve computational efficiency, ESCN  proposes to reduce the SO(3) tensor product into SO(2) in this case that the SO(3) tensor product can be equivalently replaced by a SO(2) Linear. This approximation reduces the complexity to $O(L3)$, enabling substantially faster training and inference while preserving essential symmetry properties, as demonstrated in EquiformerV2. Motivated by the efficiency gains achieved through SO(2)-based operations, in this work we further generalize the SO(2) operations beyond SO(2)-linear layers. We show that any SO(2) operations can be incorporated within these SO(2) local frames while maintaining global SO(3) equivariance, enabled by a minimal frame-averaging mechanism. In addition, these SO(2) local frames can be constructed not only on edges but also on nodes or other locations where a well-defined directional reference can be established.

---

> ### Author Response · Authors · 2025-11-28
> **Reply to reviewer b21d (2/3)**
>
> W3. Thanks for pointing it out. We wish to clarify that we compare the experimental results of adding these components and simply removing these components (the blocks are skipped). And we add notes to denote that in the revised paper. Previous ablations demonstrate that introduction of these components (SO(2) TP and SO(2) FFN) can help improve the performance compared to without these components.
> Thanks for the suggestions regarding alternative mechanisms for model design. We have included the corresponding experimental results below. Additional ablations on the SO(2) tensor product parameters $(v, M)$, the re-projection strategy, node frame construction, and frame averaging are also provided. We observe that increasing $v$ in the SO(2) tensor product consistently improves performance, with only a marginal increase in training cost. For the keep-in-frame strategy, performance improves slightly, with an associated computational overhead. This may still be a viable option for performance enhancement—potentially preferable to adjusting the cutoff or separating off-diagonal terms if  our interpretation of the keep-in-frame mechanism is the same as yours. In contrast, the continuous variant nearly doubles the training cost, making it impractical for real-world usage.
>
> ## SO(2) Tensor Product Comparison
> | Method                              | No SO(2) TP | SO(2) TP (v=1, M) | SO(2) TP (v=2, M) |
> |-------------------------------------|-------------|--------------------|--------------------|
> | MAE on Hamiltonian [10⁻⁶ Eₕ] ↓    | 36.50       | 32.86              | 31.50              |
> | Training time (sec / 100 batches) | 69          | 71                 | 72                 |
>
>
> ## Frame Handling: Re-project vs Keep-in-frame
> | Method                               | Re-project | Keep-in-frame |
> |------------|------------|----------------|
> | MAE on Hamiltonian [10⁻⁶ Eₕ] ↓    | 32.86      | 29.17          |
> | Training time (sec / 100 batches)| 72         | 79             |
>
>
> ## Node Frame Construction Methods
> | Method                                 | Node frame with averaging | Node frame with nearest-neighbor |
> |-------------|----------------|-------------------|
> | Performance [10⁻⁶ Eₕ] ↓                  | 43.62   | 31.50                            |
> | Training time (sec / 100 batches)  | 131                        | 70                               |
>
> Q1. Thanks for pointing out this question. This is a good point. The effect on training stability is difficult to quantify because there is no standard metric to directly measure such influence. To better understand the issue, we designed an experiment where we introduce Gaussian noise into the real 3D atomic positions in QM9 and examine how often the nearest-neighbor relationships between atoms flip under different noise levels.
> As shown in the table, increasing the noise level leads to a higher frequency of neighbor-flipping events. However, for the purpose of evaluating practical continuous behaviors, we focus on relatively small noise levels. When the noise magnitude is below 0.02 Bohr, the flipping ratio remains under 10%. Although some molecules still experience such changes, we believe this trade-off is reasonable compared to the computational overhead of adopting an averaged neighbor model. Note that the distance between H and H in $H_2$ is about 1.4 Bhor.
>
> | Sigma (Bohr) | Mean Change Rate | Std Dev |
> |--------------|------------------|---------|
> | 0.0010       | 0.0236           | 0.0344  |
> | 0.0050       | 0.0656           | 0.0485  |
> | 0.0100       | 0.0853           | 0.0544  |
> | 0.0200       | 0.0944           | 0.0596  |
> | 0.0500       | 0.1121           | 0.0691  |
> | 0.1000       | 0.1307           | 0.0729  |
> | 0.2000       | 0.1482           | 0.0782  |
> | 0.5000       | 0.2876           | 0.0984  |

---

> ### Author Response · Authors · 2025-11-28
> **Reply to reviewer b21d (3/3)**
>
> Q2. For sure. We wish to point out that the SO(2) TP operations are $O(M^2)$ and for the rotation, the complexity is $O(L^3)$. And we add this in the revised paper to avoid confusion. Thanks for pointing it out. The computational cost for simple SO(2) TP between two SO(2) irreps will be $O(M^2)$, where $M$ is the cutoff of the order on the SO(2) irreps. In our implementation, we set $M=L$, since the SO(2) irreps are transferred from the SO(3) irreps whose cutoff is $L$. To see why the time complexity is linear in $M$, consider the computation for a single channel.  The SO(2) TP enumerates all valid index pairs $(m1,m2)$ that satisfy the SO(2) selection rule $m=m1+m2$ and $m =|m_1 - m_2|$. Because both $m_1$​ and $m_2$​ range from $0$ to $M$, the number of valid combinations is proportional to $M^2$. For each valid path, the computational work is $O(1)$, consisting of a single complex multiplication between two SO(2) feature components. Therefore, the total complexity will be $M^2$ for two SO(2) irreps as input of SO(2) TP. Note that the time complexity of rotating into the SO(2) local frame and rotating back is $O(L_{max}^3)$. This is because multiplying an irreducible representation of degree $\ell$ by the corresponding rotation matrix requires $\ell^2$ operations. Summing this cost over all degrees $\ell = 0, 1, \ldots, L_{max}$ yields a total complexity proportional to $\sum_{\ell=0}^{L_{max}} \ell^2 = O(L_{max}^3)$.
>
> Q3. Thanks for your suggestions. Here is the profile for the ConvLayer, SO(2) TP, FFN, gating, and LayerNorm. The batch size is set to 32, and the reported results for each example. You will find that Nodewise Interaction and SO(2) FFN takes the majority part of inference time.
> | Component Type  | Count | Total Time (ms) | Memory (GB) |
> |------------------------|-------|------------------------|------------------|
> | Nodewise Interaction   | 3     | 81.9049                | 0.432772         |
> | SO2 FFN                      | 3     | 22.6721                | 0.286858         |
> | SO2_TP                        | 3     | 15.5794                | 0.031491         |
> | SO(2) Gate                   | 9     | 14.4533                | 0.196731         |
> | SO(2) LayerNorm           | 3     | 3.7635                 | 0.007032         |

---

### Author Response · Authors · 2025-11-28
**Letter to reviewers**

Dear reviewers,

Thank you for taking the time to read our manuscript and for providing valuable feedback. We have completed several additional experiments and posted the corresponding rebuttal content. We apologize for the late response, as these experiments required substantial runtime, and we appreciate your understanding. Thank you again for your efforts and engagement throughout the rebuttal process, and we hope our responses can somehow alleviate your concerns.

Best,

Authors.

---

### Meta-Review · Area_Chair_s1Hw · 2026-01-06

**Summary:**

Despite the reported efficiency gains, the paper fails to meet the ICLR bar for methodological innovation. The decision to reject is based on the consensus among reviewers regarding the incremental nature of the work and the authors' failure to adequately distinguish their contribution from established baselines during the rebuttal. The core technical premise—achieving computational efficiency by reducing $SO(3)$ operations to $SO(2)$ using local frames—is not a novel contribution of this paper. This methodology is well-established in the literature, most notably by eSCN, EquiformerV2, and DeepH-2. As explicitly noted by three reviewers (mpT8, gAzZ, and cJL7), the proposed approach apply these mature techniques to the task of Hamiltonian prediction. Extending $SO(2)$-linear layers to other standard blocks (Gates, LayerNorms, FFNs) is viewed as engineering exercise rather than a significant algorithmic advancement. When explicitly asked by multiple reviewers to provide a "clearer articulation of what new theoretical or algorithmic insights distinguish it from existing SO(2)-based approaches," the authors failed to provide a substantive answer. Instead, they relied on vague, repetitive statements (e.g., "we further generalize the SO(2) operations beyond SO(2) linear layers") without identifying a unique theoretical mechanism or distinct insight. The inability to clearly differentiate the work from eSCN suggests the contribution is somewhat limited.

**Reviewer Concerns:**

The main concern is regarding the incremental novelty compared to existing SO(3) to SO(2) baselines. The authors failed to address it in a convincing way.

**Reviewer Scores:**

The first reviewer b21d might increase the score to 6. I don't believe the remaining reviewers will increase the scores, as their core concern, the limited methodological novelty, is not addressed.

---

### Decision · Program_Chairs · 2026-01-26

Reject